# Brain and molecular mechanisms underlying the nonlinear association between close friendships, mental health, and cognition in children

Chun Shen[1,2,3†], Edmund T Rolls[1,4,5†], Shitong Xiang[1,2], Christelle Langley[6,7], Barbara J Sahakian[1,6,7], Wei Cheng[1,2,8,9,10]*, Jianfeng Feng[1,2,3,4,8,11]*

[1]Institute of Science and Technology for Brain-Inspired Intelligence, Fudan University, Shanghai, China; [2]Key Laboratory of Computational Neuroscience and Brain-Inspired Intelligence (Fudan University), Ministry of Education, Shanghai, China; [3]State Key Laboratory of Medical Neurobiology and MOE Frontiers Center for Brain Science (Fudan University), Ministry of Education, Shanghai, China; [4]Department of Computer Science, University of Warwick, Coventry, United Kingdom; [5]Oxford Centre for Computational Neuroscience, Oxford, United Kingdom; [6]Department of Psychiatry, University of Cambridge, Cambridge, United Kingdom; [7]Behavioural and Clinical Neuroscience Institute, University of Cambridge, Cambridge, United Kingdom; [8]Fudan ISTBI—ZJNU Algorithm Centre for Brain-Inspired Intelligence, Zhejiang Normal University, Jinhua, China; [9]Shanghai Medical College and Zhongshan Hospital Immunotherapy Technology Transfer Center, Shanghai, China; [10]Department of Neurology, Huashan Hospital, Fudan University, Shanghai, China; [11]Zhangjiang Fudan International Innovation Center, Shanghai, China

*For correspondence:
wcheng.fdu@gmail.com (WC);
jffeng@fudan.edu.cn (JF)

†These authors contributed equally to this work

**Abstract** Close friendships are important for mental health and cognition in late childhood. However, whether the more close friends the better, and the underlying neurobiological mechanisms are unknown. Using the Adolescent Brain Cognitive Developmental study, we identified nonlinear associations between the number of close friends, mental health, cognition, and brain structure. Although few close friends were associated with poor mental health, low cognitive functions, and small areas of the social brain (e.g., the orbitofrontal cortex, the anterior cingulate cortex, the anterior insula, and the temporoparietal junction), increasing the number of close friends beyond a level (around 5) was no longer associated with better mental health and larger cortical areas, and was even related to lower cognition. In children having no more than five close friends, the cortical areas related to the number of close friends revealed correlations with the density of μ-opioid receptors and the expression of OPRM1 and OPRK1 genes, and could partly mediate the association between the number of close friends, attention-deficit/hyperactivity disorder (ADHD) symptoms, and crystalized intelligence. Longitudinal analyses showed that both too few and too many close friends at baseline were associated with more ADHD symptoms and lower crystalized intelligence 2 y later. Additionally, we found that friendship network size was nonlinearly associated with well-being and academic performance in an independent social network dataset of middle-school students. These findings challenge the traditional idea of 'the more, the better,' and provide insights into potential brain and molecular mechanisms.

## Editor's evaluation

The findings of this study yield important new insights into the relationship between the number of close friends and mental health, cognition, and brain structure. Due to the large sample sizes, the evidence is solid but would have been improved if both of the analysed datasets contained more closely matched measures. This work advances our understanding of how the friendship network relates to young adolescents' mental well-being and cognitive functioning and their underlying neural mechanisms.

## Introduction

Late childhood and its transition toward adolescence is a period marked by decreasing parental influence alongside increasing peer influence. It is a period critical for social interaction, during which friendships are especially important (*Blakemore and Mills, 2014*). During this period, the social brain is still undergoing significant development, in parallel with changes in social cognition (*Mills et al., 2014*). Meanwhile, evidence suggests that psychiatric disorders often have an onset in adolescence (*Kessler et al., 2005*), which may be partly influenced by the concurrent changes in the social environment and brain (*Paus et al., 2008*). Therefore, understanding the relationship between friendship, mental health, and cognition during this period, and the underlying brain mechanisms, is of considerable clinical and public health importance.

It has been well established that positive social relationships such as close friendships are essential for mental health and cognition in children and adolescents (*Marion et al., 2013*; *Narr et al., 2019*; *Wentzel et al., 2018*). However, it remains unclear whether having more close friends is necessarily better. Cognitive constraints and time resources limit the number of close social ties that an individual can maintain simultaneously (*Dunbar, 2018*). The innermost layer of the friendship group with the highest emotional closeness is around five close friends (the so-called Dunbar's number) (*Zhou et al., 2005*). For now, only a few empirical studies have examined the nonlinear association between social relationships, mental health, and cognition in children and adolescents. For instance, a large study of a nationally representative sample in the United States reported that adolescents with either too many or too few friends had higher levels of depressive symptoms (*Falci and McNeely, 2009*). Two large-scale studies reported that the benefits of social interactions for well-being were nearly negligible once the quantity reached a moderate level (*Kushlev et al., 2018*; *Ren et al., 2022*). Additionally, a significant U-shaped effect was detected between positive relations with others and cognitive performance (*Brown et al., 2021*). Overall, the assumption of linearity still dominates studies of social relationships, and the effect of the friendship network size at the high end remains largely unexplored.

Despite a large body of evidence linking friendships to mental health and cognition, we know relatively little about the underlying mechanisms involved (*Pfeifer and Allen, 2021*). The social brain hypothesis proposes that the evolution of brain size is driven by complex social selection pressures (*Dunbar and Shultz, 2007*). Animal studies have shown that social network size can predict the volume of the mid-superior temporal sulcus (*Sallet et al., 2011*; *Testard et al., 2022*), a region in which neurons respond to socially relevant stimuli such as face expression and head movement to make or break social contact (*Hasselmo et al., 1989a*; *Hasselmo et al., 1989b*). In human neuroimaging studies, several key brain regions, including the medial prefrontal cortex (mPFC, i.e. orbitofrontal [OFC] and anterior cingulate cortex [ACC]), the cortex in the superior temporal sulcus (STS), the temporoparietal junction (TPJ), amygdala, and the anterior insula, have been implicated in social cognitive processes (*Frith and Frith, 2007*). Moreover, there has been an increasing number of studies dedicated to investigating the social brain in children and adolescents over the past decade (*Andrews et al., 2021*; *Burnett et al., 2011*).

At the molecular level, the μ-opioid receptor is widely distributed in the brain, particularly in regions associated with social pain such as the ACC and anterior insula (*Baumgärtner et al., 2006*). Recent studies have identified the crucial role of μ-opioid receptors in forming and maintaining friendships (*Dunbar, 2018*), and variations in the μ-opioid receptor gene have been related to individual differences in rejection sensitivity (*Way et al., 2009*). In addition, other neurotransmitters, including dopamine, serotonin, GABA, and noradrenaline, may interact with the opioids, and are involved in social affiliation and social behavior (*Machin and Dunbar, 2011*). Dysregulation of the social brain and neurotransmitter systems is also implicated in the pathophysiology of major psychiatric disorders (*Porcelli et al., 2019*). Taken together, it is suggested that changes

**eLife digest** Close friendships are crucial during the transition from late childhood to adolescence as children become more independent from their parents and influenced by their peers. The brain undergoes a tremendous amount of development during this period, and it is also a time when mental health disorders often begin to emerge.

Scientists are still learning about how friendships shape brain development and mental health during this transition. Maintaining friendships takes time and mental resources so there may be limits on how many friends are beneficial. Here, Shen, Rolls et al. show that the having more friends is not always directly related to better mental health and cognitive abilities.

In the study, Shen, Rolls et al. analyzed data from nearly 7,500 young people between around 10 to 12 years old: this included, their number of close friends, their mental health and cognitive abilities such as working memory, attention and processing speed, and images of their brains. Data from a second set of about 16,000 young people were then analyzed to confirm the results.

Shen, Rolls et al. found having a higher number of close friends was associated with improved mental health and cognitive ability. However, this association stopped once around five friends had been reached, after which having more friends was no longer linked to better mental health and was even correlated with lower cognition. Additionally, individuals with too few or too many friends had more symptoms of Attention-deficit/hyperactivity disorder (ADHD) and were less able to learn from their experiences.

This non-linear relationship between number of friends and mental health and cognitive abilities can be partly explained by the structure of the brain. Shen, Rolls et al. found that brain regions associated with friendship were larger in individuals with more close friends, but did not increase any further once the number of friends a person had exceeded five individuals with around five close friends also had more of a receptor that is part of the opioid system, which may make them more responsive to laughter, friendly touch, or other positive social interactions.

These findings challenge the idea that having more friends is always better. It also provides insights into how friendships affect brain health during the transition from late childhood to adolescence. Insights from this study may aid the development of interventions to support healthy brain development during youth.

in the social brain might explain the relationship between social connections and mental health (*Lamblin et al., 2017*). However, the empirical evidence on this topic is limited in late childhood and adolescence.

In this study, we aimed to investigate the relationship between the number of close friends, mental health, and cognitive outcomes, with a focus on potential nonlinear associations. We used data from the Adolescent Brain Cognitive Developmental (ABCD) study (*Karcher and Barch, 2021*) and an independent social network dataset (*Paluck et al., 2016*). These datasets provided reliable measures of close friend quantity, mental health, and cognition, and included a combined total of more than 23,000 participants (*Figure 1a*). To evaluate the potential nonlinear relation between friendship quantity (predictor) and mental health and cognition (outcome), two different analytic approaches were utilized. Specifically, we examined the presence of a significant quadratic term as an indicator of nonlinearity, and subsequently conducted a two-lines test (*Simonsohn, 2018*) to estimate an interrupted regression and identify the breakpoint (*Figure 1b*). To explore the underlying neurobiological mechanisms, we further tested the nonlinear association between the number of close friends and brain structure. We then correlated the related brain differences with the density of eight neurotransmitter systems, as well as the expression of the μ-opioid receptor gene (OPRM1) and the κ-opioid receptor gene (OPRK1) (*Figure 1c*). Finally, longitudinal and mediation analyses were conducted to uncover the direction and direct association between the number of close friends, mental health, cognition, and brain structure (*Figure 1d*). Based on the existing literature, we hypothesized that the number of close friends was nonlinearly related to mental health, cognition, and the social brain; and that this relationship could potentially be explained by brain differences and molecular mechanisms.

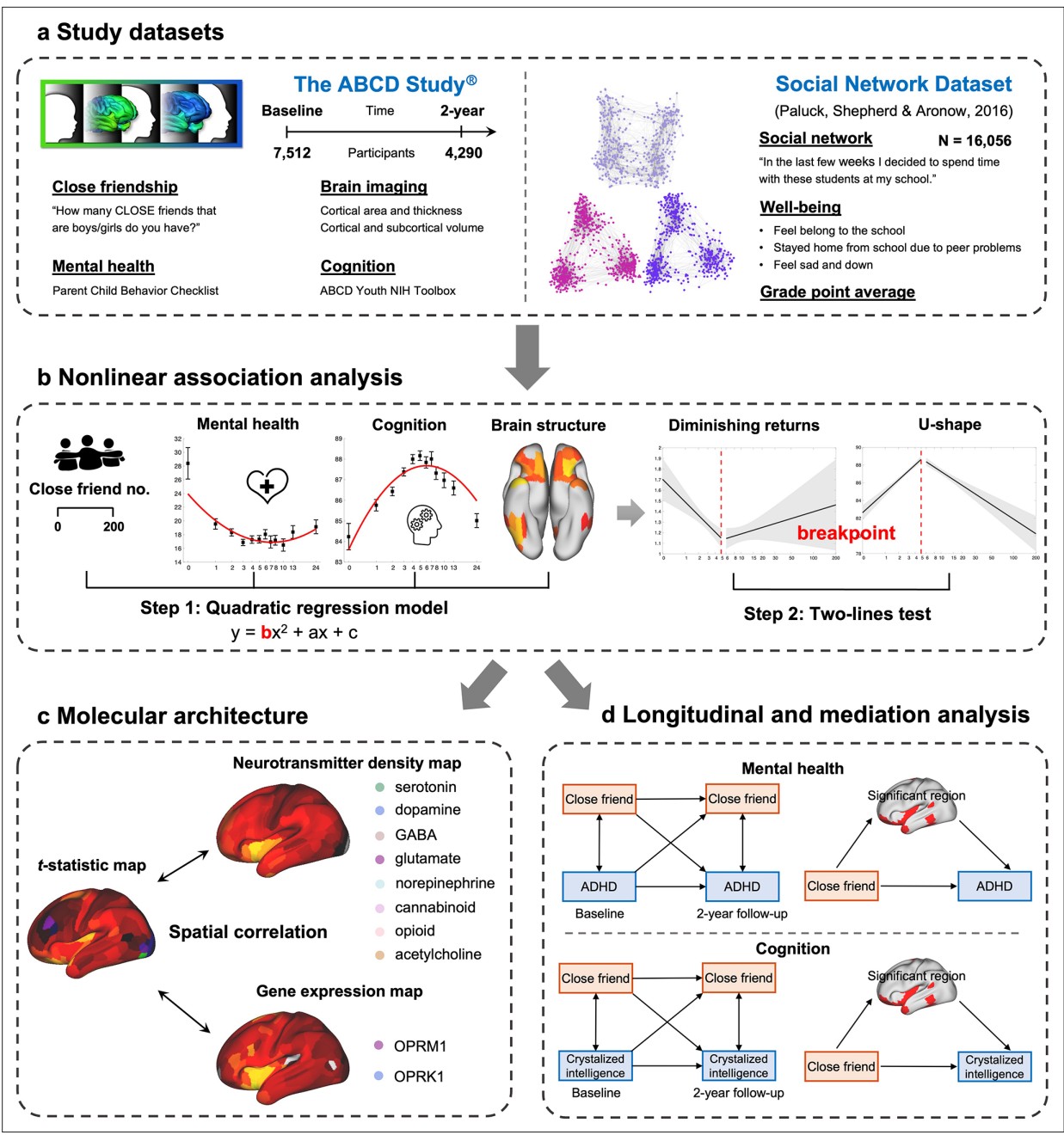

**Figure 1.** The study workflow. (**a**) Study datasets and key measures used in the present study. (**b**) A two-step approach to evaluate the nonlinear association. The number of close friends is used as the independent variable in quadratic regression models. Once a significant squared term ('b') is found, a two-lines test is conducted to estimate the breakpoint. Then participants are classified into two groups according to the breakpoint. (**c**) Correlation of brain differences related to the number of close friends with neurotransmitter density and gene expression level. (**d**) Longitudinal and mediation analysis of the number of close friends, ADHD symptoms, crystalized intelligence, and the significant surface areas.

## Results

### Demographic characteristics

In the ABCD study, 7512 participants (3625 [48.3%] females, aged 9.91 ± 0.62 y) provided self-reported number of close friends, a broad range of mental health and cognitive measures, and quality-controlled MRI data at baseline (*Table 1*), and 4290 of them (2044 [47.7%] females, aged 11.49 ± 0.66 y) had 2-year follow-up data available (*Table 2*). In the social network dataset, 16,065 subjects

**Table 1.** Characteristics of the study population in the Adolescent Brain Cognitive Developmental (ABCD) study at baseline*.

| | ≤5 close friends (N = 4863) | >5 close friends (N = 2649) | p-value[†] |
|---|---|---|---|
| Age | 9.91 ± 0.61 | 9.93 ± 0.63 | 0.08 |
| Sex | | | 0.02 |
| Female | 2299 (47.3%) | 1326 (50.1%) | |
| Male | 2564 (52.7%) | 1323 (49.9%) | |
| Race | | | 0.001 |
| White | 2672 (54,9%) | 1497 (56.5%) | |
| Black | 561 (11,5%) | 370 (14.0%) | |
| Hispanic | 1015 (20.9%) | 485 (18.3%) | |
| Asian | 109 (2.2%) | 45 (1.7%) | |
| Other | 506 (10.4%) | 252 (9.5%) | |
| Family size [‡] | 4.69 ± 1.82 | 4.58 ± 1.84 | 0.01 |
| Family income | 7.28 ± 2.34 | 7.4 ± 2.35 | 0.03 |
| Parental education | 16.82 ± 2.62 | 16.97 ± 2.53 | 0.02 |
| Body mass index | 18.71 ± 4.11 | 18.73 ± 4.11 | 0.77 |
| Puberty | 1.73 ± 0.86 | 1.78 ± 0.88 | 0.02 |
| Urbanization [§] | | | 0.12 |
| Rural | 395 (8.5%) | 220 (8.7%) | |
| Urban clusters | 167 (3.6%) | 68 (2.7%) | |
| Urbanized area | 4,074 (87.9%) | 2,229 (88.6%) | |
| Total close friends | 3.04 ± 1.36 | 12.19 ± 13.14 | <0.001 |
| Same-sex close friends | 2.41 ± 1.22 | 9.13 ± 9.82 | <0.001 |
| Opposite-sex close friends | 0.63 ± 0.79 | 3.05 ± 5.79 | <0.001 |

*Values are mean ± SD or N (%).
[†]For continuous data, *t*-test was performed; for categorical data, chi-square test was performed.
[‡]4780 and 2624 participants in ≤5 and >5 close friends groups have family size data, respectively.
[§]4636 and 2517 participants in ≤5 and >5 close friends groups have urbanization data, respectively.

from 48 middle schools (8065 [50.3%] female, aged 12.00 ± 1.03 y) who had complete key variables were included (*Table 3*).

## Nonlinear association between the number of close friends, mental health, and cognition

The number of close friends was significantly associated with 12 out of 20 mental health measures, and 7 out of 10 cognitive scores at baseline (the total F-value of the linear and quadratic terms, p<0.05/30; *Figure 2a–g*). For these 18 outcomes except the withdrawn/depressed, all quadratic terms reached significance after Bonferroni corrections (p<0.05/60), and all quadratic models provided a significantly better fit than the corresponding linear models (F = [13.25, 55.53], all p<0.001). For mental health, the greatest effect sizes of the quadratic terms were observed for social problems ($\beta$ = 0.08, t = 5.92, p=3.3 × 10$^{-9}$, $\Delta R^2$ = 0.43%) and attention problems ($\beta$ = 0.12, t = 5.83, p = 5.8 × 10$^{-9}$, $\Delta R^2$ = 0.42%), suggesting that the quadratic term of close friend quantity additionally explained 0.43 and 0.50% of the variability compared with the corresponding linear model. For cognition, the greatest effect sizes of the quadratic terms were observed for total intelligence ($\beta$ = –0.35, t = –7.45, p=1.0 × 10$^{-13}$, $\Delta R^2$ = 0.50%) and crystalized intelligence ($\beta$ = –0.26, t = –6.87, p=6.7 × 10$^{-12}$, $\Delta R^2$ = 0.43%)

**Table 2.** Characteristics of the study population in the Adolescent Brain Cognitive Developmental (ABCD) study at 2-year follow-up (N = 4290).

|  | Value* |
| --- | --- |
| Age | 11.49 ± 0.66 |
| Sex |  |
| Female | 2044 (47.7%) |
| Male | 2246 (52.4%) |
| Race |  |
| White | 2612 (60.9%) |
| Black | 385 (9.0%) |
| Hispanic | 791 (18.4%) |
| Asian | 89 (2.1%) |
| Other | 413 (9.6%) |
| Family income | 7.83 ± 2.04 |
| Parental education | 17.11 ± 2.44 |
| Body mass index | 20.35 ± 4.63 |
| Puberty | 2.53 ± 1.05 |
| Total close friends | 6.82 ± 8.37 |
| Same-sex close friends | 4.99 ± 5.92 |
| Opposite-sex close friends | 1.83 ± 3.66 |

*Values are mean ± SD or N (%).

**Table 3.** Characteristics of the study population in the social network dataset (N = 16,056).

| Variable | Value * |
| --- | --- |
| Age | 12.00 ± 1.03 |
| Sex |  |
| Female | 8068 (50.3%) |
| Male | 7988 (49.8%) |
| Grade |  |
| 5th grade | 1107 (6.9%) |
| 6th grade | 4190 (26.1%) |
| 7th grade | 5279 (32.9%) |
| 8th grade | 5480 (34.1%) |
| New to the school |  |
| New to school | 4315 (26.9%) |
| Returning to school | 11,741 (73.1%) |
| Most friends go to this school |  |
| Yes | 14,429 (89.9%) |
| No | 1627 (10.1%) |
| Outdegree | 8.08 ± 2.43 |
| Indegree | 7.83 ± 4.42 |
| Reciprocal degree | 3.82 ± 2.14 |
| Well-being | 0.86 ± 0.24 |
| Grade point average | 3.17 ± 0.61 |

*Values are mean ± SD or N (%).

(*Figure 2—figure supplement 1*). The findings were robust with respect to random choice of the siblings (*Figure 2—figure supplement 2*).

The average breakpoint of the number of close friends for the mental health and cognitive outcomes with significant quadratic terms was 4.89 ± 0.68 (*Figure 2h*). Both mental health and cognition were positively associated with close friend quantity, with an ideal number of around 5. These nonlinear associations were consistent in males and females (*Figure 2—figure supplement 3*). However, the number of same-sex close friends, but not of opposite-sex close friends, was significantly related to mental health and cognition (26 out of 30 measures with a significant $F$ value after Bonferroni correction), and children with 4.05 ± 0.59 same-sex close friends had the best mental health and cognitive functions (*Figure 2—figure supplement 4*).

Finally, the same analyses were performed using the cross-sectional data collected at 2 y later (*Figure 2—figure supplement 5*). The number of close friends was significantly associated with 10 out of 20 mental health measures, and 3 out of 6 cognitive scores. Significant nonlinear associations were observed between close friend quantity and five measures, with an average breakpoint of 4.60 ± 0.55 close friends. The greatest effect sizes of the quadratic terms were observed for attention problems ($\beta$ = 0.10, $t$ = 3.63, p=2.9 × 10$^{-4}$, $\Delta R^2$ = 0.27%) and crystalized intelligence ($\beta$ = –0.24, $t$ = –4.70, p=2.7 × 10$^{-6}$, $\Delta R^2$ = 0.36%) for mental health and cognition, respectively.

## The number of close friends was quadratically associated with brain structure

In the ABCD study, the number of close friends was significantly associated with the total cortical area ($F$ = 6.29, p=1.0 × 10$^{-3}$; *Figure 3d*), and the total cortical volume ($F$ = 5.80, p=3.1 × 10$^{-3}$). No

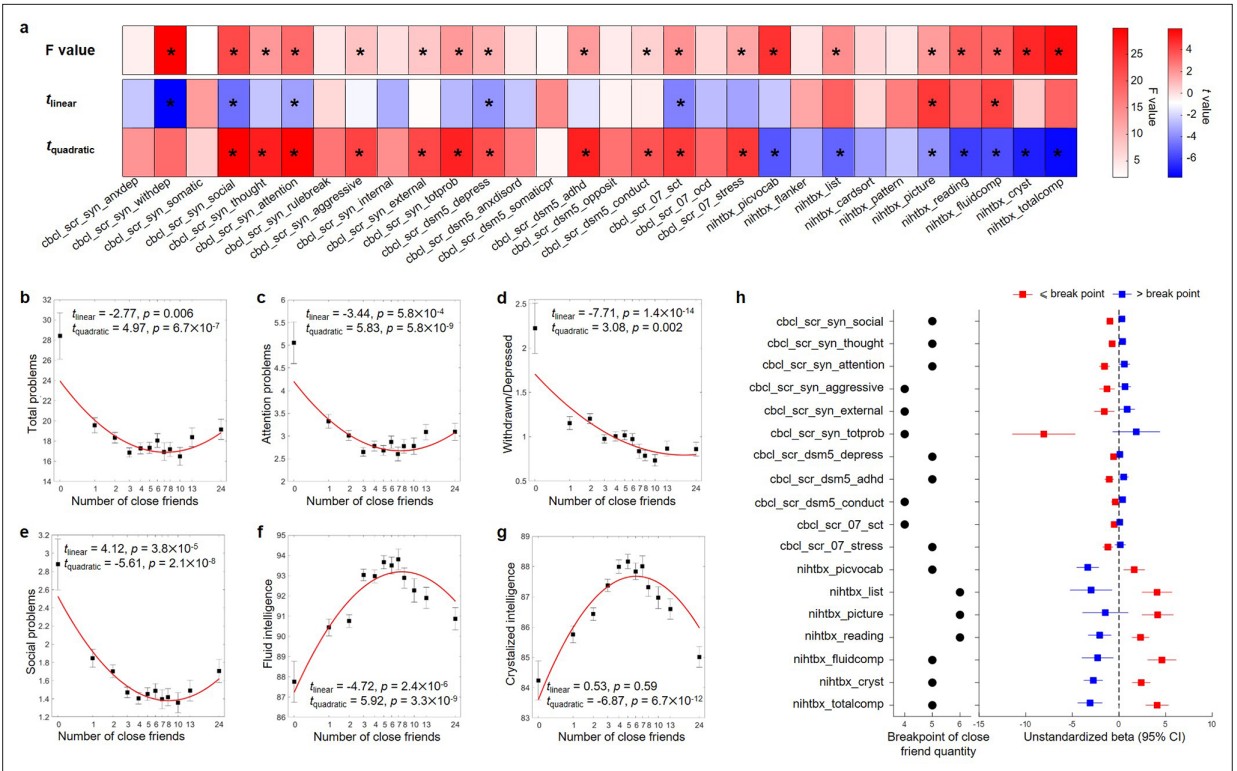

**Figure 2.** Results of behavior-level nonlinear association analyses in the Adolescent Brain Cognitive Developmental (ABCD) study at baseline. (**a**) Results of quadratic regression models. The total $F$ values of quadratic and linear terms, and the $t$ values of linear and quadratic terms are reported. An asterisk indicates statistical significance after Bonferroni correction (i.e., $p<0.05/30$ for $F$ value, and $p<0.05/60$ for $t$ value). Relationship between the number of close friends and the total problems (**b**), attention problems (**c**), withdrawn/depressed (**d**), social problems (**e**), fluid intelligence (**f**), and crystalized intelligence (**g**). The number of close friends is classified into 13 bins, sample sizes of which are 107, 585, 1104, 1196, 957, 914, 631, 399, 463, 363, 416 and 477. In each bin, the mean (i.e., black dot) and standard error (i.e., error bar) of the dependent variable are shown. The x-axis is in log scale, and the median of the number of close friends in each bin was labeled in the x-axis. The red line is the fitted quadratic model. (**h**) Results of the two-lines tests. The breakpoint and the estimated coefficients with 95% confidence intervals of linear regressions in each group separated by the breakpoint are reported. cbcl-scr-syn-anxdep, Anxious/Depressed Syndrome Scale; cbcl-scr-syn-withdep, Withdrawn/Depressed Syndrome Scale; cbcl-scr-syn-somatic, Somatic Complaints Syndrome Scale; cbcl-scr-syn-social, Social Problems Syndrome Scale; cbcl-scr-syn-thought, Thought Problems Syndrome Scale; cbcl-scr-syn-attention, Attention Problems Syndrome Scale; cbcl-scr-syn-rulebreak, Rule-Breaking Behavior Syndrome Scale; cbcl-scr-syn-aggressive, Aggressive Behavior Syndrome Scale; cbcl-scr-syn-internal, Internalizing Problems Syndrome Scale; cbcl-scr-syn-external, Externalizing Problems Syndrome Scale; cbcl-scr-syn-totprob, Total Problems Syndrome Scale; cbcl-scr-dsm5-depress, Depressive Problems DSM-5 Scale; cbcl-scr-dsm5-anxdisord, Anxiety Problems DSM-5 Scale; cbcl-scr-dsm5-somaticpr, Somatic Problems DSM-5 Scale; cbcl-scr-dsm5-adhd, ADHD DSM-5 Scale; cbcl-scr-dsm5-opposite, Oppositional Defiant Problems DSM-5 Scale; cbcl-scr-dsm5-conduct, Conduct Problems DSM-5 Scale; cbcl-scr-07-sct, Sluggish Cognitive Tempo Scale2007 Scale; cbcl-scr-07-ocd, Obsessive-Compulsive Problems Scale2007 Scale; cbcl-scr-07-stress, Stress Problems Scale2007 Scale; nihtbx-picvocab, Picture Vocabulary Test; nihtbx-flanker, Flanker Inhibitory Control and Attention Test; nihtbx-list, List Sorting Working Memory Test; nihtbx-cardsort, Dimensional Change Card Sort Test; nihtbx-pattern, Pattern Comparison Processing Speed Test; nihtbx-picture, Picture Sequence Memory Test; nihtbx-reading, Oral Reading Recognition Test; nihtbx-fluidcomp, Fluid Composite Score; nihtbx-cryst, Crystallized Composite Score; nihtbx-totalcomp, Total Composite Score.

The online version of this article includes the following source data and figure supplement(s) for figure 2:

**Source data 1.** Results of behavior-level nonlinear association analyses in the Adolescent Brain Cognitive Developmental (ABCD) study at baseline.

**Figure supplement 1.** Effect sizes of linear and quadratic terms of close friend number in the Adolescent Brain Cognitive Developmental (ABCD) study at baseline.

**Figure supplement 2.** Behavior-level results of quadratic regression models by random choice of the siblings in the Adolescent Brain Cognitive Developmental (ABCD) study at baseline.

**Figure supplement 3.** Results of behavior-level nonlinear association analyses in the Adolescent Brain Cognitive Developmental (ABCD) study at baseline in girls and boys, respectively.

**Figure supplement 4.** Nonlinear association of the number of same-sex and opposite-sex close friends with mental health and cognition in the Adolescent Brain Cognitive Developmental (ABCD) study at baseline.

*Figure 2 continued on next page*

*Figure 2 continued*

**Figure supplement 5.** Results of behavior-level nonlinear association analyses in the Adolescent Brain Cognitive Developmental (ABCD) study at 2-year follow-up.

significant relationship between the number of close friends and mean cortical thickness ($F = 0.62$, p=0.54) and total subcortical volume ($F = 3.94$, p=0.02) was found.

After false discovery rate (FDR) correction ($q < 0.05$), the significant cortical areas associated with the number of close friends were mainly located in the OFC, insula, the ACC, the anterior temporal cortex, and the TPJ (**Figure 3a**). The brain region with the largest effect size for the linear term was the OFC (left medial OFC [area 11l and 13l] and lateral OFC [area 47m and 47s]). The quadratic terms of the number of close friends for all these regions were significant (**Figure 3b**), and the greatest effect sizes were observed in the temporal pole (left STGa: $\beta = -0.58$, $t = -4.20$, p=$2.8 \times 10^{-5}$, $\Delta R^2 = 0.18\%$, **Figure 3e**; right TGd: $\beta = -3.32$, $t = -4.11$, p=$4.0 \times 10^{-5}$, $\Delta R^2 = 0.18\%$, **Figure 3f**). These findings were robust for random choice of the siblings (**Figure 3—figure supplement 1**). Similar findings were found for cortical volumes (**Figure 3—figure supplement 2**). As the correlation of cortical area and

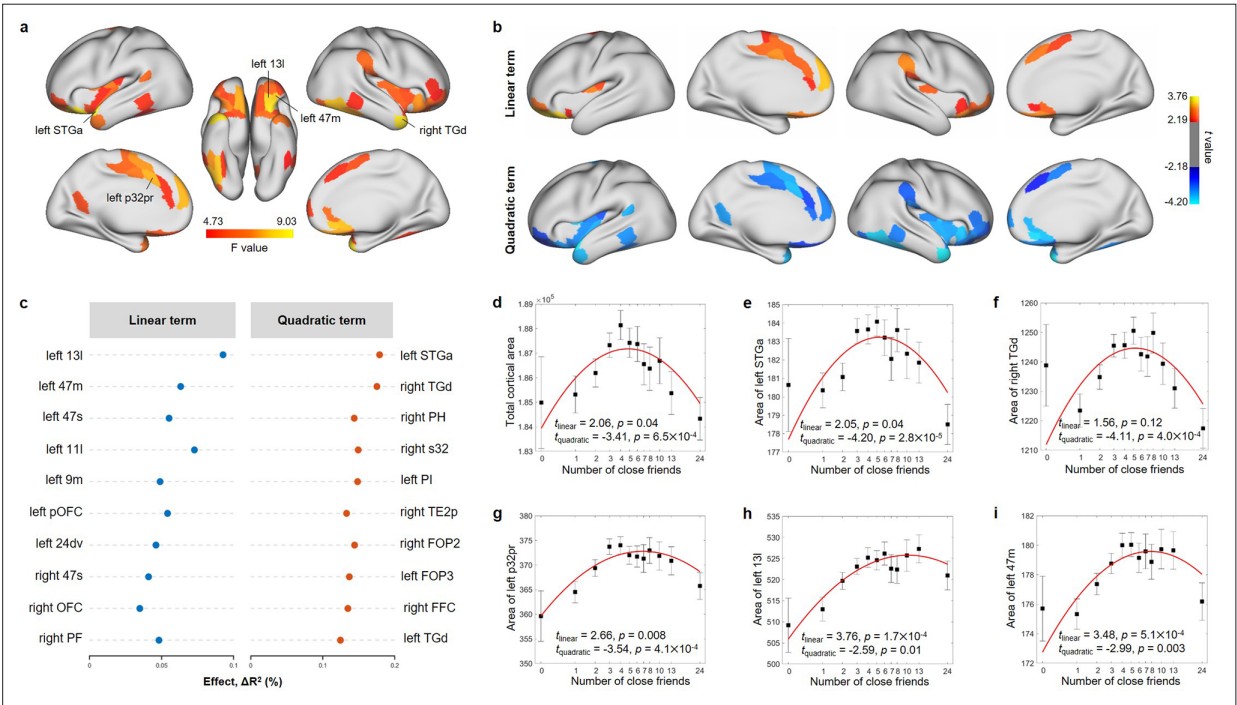

**Figure 3.** Nonlinear association between the number of close friends and cortical area in the Adolescent Brain Cognitive Developmental (ABCD) study at baseline. (**a**) Cortical areas significantly associated with the number of close friends after FDR correction (i.e., 360 regions) based on the total $F$ values of linear and quadratic terms. (**b**) Cortical areas with a significant linear or quadratic term. FDR correction was performed within the significant regions obtained in (**a**). (**c**) Top 10 regions with the strongest effect sizes of linear and quadratic terms, respectively. Relationship between the number of close friends and the total cortical area (**d**), left STGa (**e**), right TGd (**f**), left p32pr (**g**), left 13l (**h**), and left 47m (**i**). The number of close friends is classified into 13 bins, sample sizes of which are 107, 585, 1104, 1196, 957, 914, 631, 399, 463, 363, 416 and 377. In each bin, the mean (i.e., black dot) and standard error (i.e., error bar) of the dependent variable are shown. The x-axis is in log scale, and the median of the number of close friends in each bin was labeled in the x-axis. The red line is the fitted quadratic model. The names of the brain regions are from the HCP-MMP atlas.

The online version of this article includes the following source data and figure supplement(s) for figure 3:

**Source data 1.** Results of nonlinear association analyses between the number of close friends and cortical area in the Adolescent Brain Cognitive Developmental (ABCD) study at baseline.

**Figure supplement 1.** Nonlinear association between the number of close friends and cortical areas by random choice of the siblings.

**Figure supplement 2.** Nonlinear association between the number of close friends and cortical volumes.

**Figure supplement 3.** Relationship between cortical area and cortical volume.

**Figure supplement 4.** Results of two-lines tests for significant cortical areas.

**Figure supplement 5.** Results of linear association analyses between close friend quantity and cortical area in ≤5 and >5 groups, respectively.

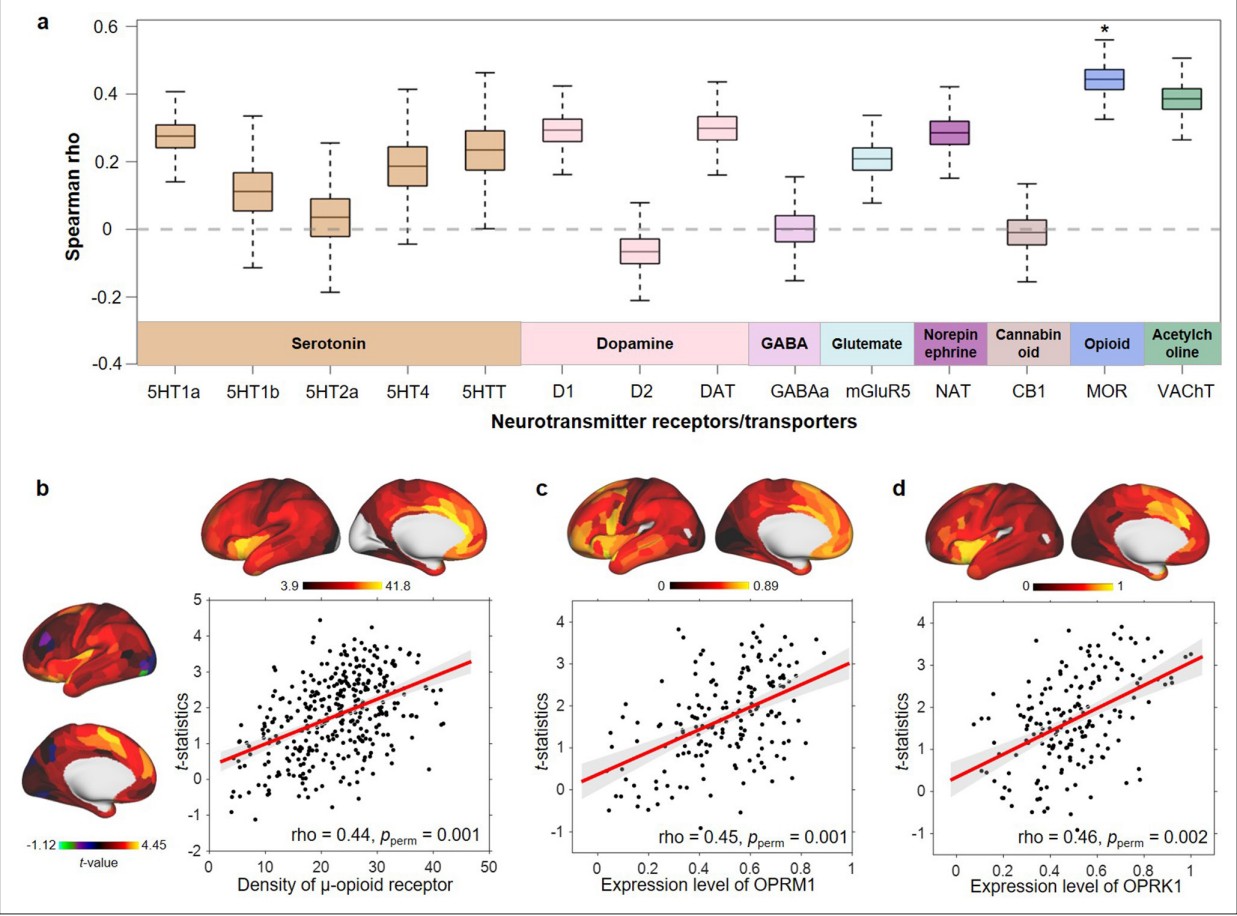

**Figure 4.** Spatial correlation between cortical area differences related to the number of close friends in children with ≤5 close friends and density of neurotransmitters and gene expression level. (**a**) Bootstrapped Spearman correlations (10,000 times) between *t*-statistics of close friendship quantity and densities of 14 neurotransmitter receptors or transporters. In each box, the line indicates the median and the whiskers indicate the 5th and 95th percentiles. p-Values were estimated by 5000 times permutation. *: Bonferroni corrected $p_{perm} < 0.05$. MOR: μ-opioid receptor. (**b**) The scatter map of *t*-statistics of close friendship quantity and the density of the μ-opioid receptor. (**c**) The scatter map of *t*-statistics of close friendship quantity and the expression level of OPRM1 gene. (**d**) The scatter map of *t*-statistics of close friendship quantity and the expression level of OPRK1 gene.

cortical volume with the number of close friends is high ($r = 0.78$, p=3.3 × 10$^{-76}$) and cortical area and volume themselves are highly correlated ($r = 0.92$, p=9.0 × 10$^{-151}$; *Figure 3—figure supplement 3*), we focused on cortical area in the following analyses.

Further, two-lines tests suggested that participants with around five close friends (breakpoint = 5.30 ± 0.85) had the largest areas in these cortical regions (*Figure 3—figure supplement 4*). To illustrate the patterns of nonlinear relationships, we performed linear regression models in participants with ≤5 and >5 close friends, respectively. Similar regions to those found with quadratic models including the OFC, insula, the ACC, and temporal cortex were significant after FDR correction in the ≤5 group (*Figure 3—figure supplement 5a and b*), and the largest effect size was observed in the OFC (*Figure 3—figure supplement 5c*). However, the number of close friends was not related to cortical area in the >5 group (*Figure 3—figure supplement 5d*). Moreover, the cortical area associative patterns of close friend quantity in the two groups were not correlated ($r = –0.02$, p=0.78; *Figure 3—figure supplement 5e*).

## Relationship to molecular architecture

As the number of close friends was nonlinearly associated with cortical area and the significant regions were only found in participants with no more than five close friends, we focused on the brain associative pattern for the number of close friends in the ≤5 group. We found that the correlations between the spatial pattern of cortical area related to the number of close friends and densities

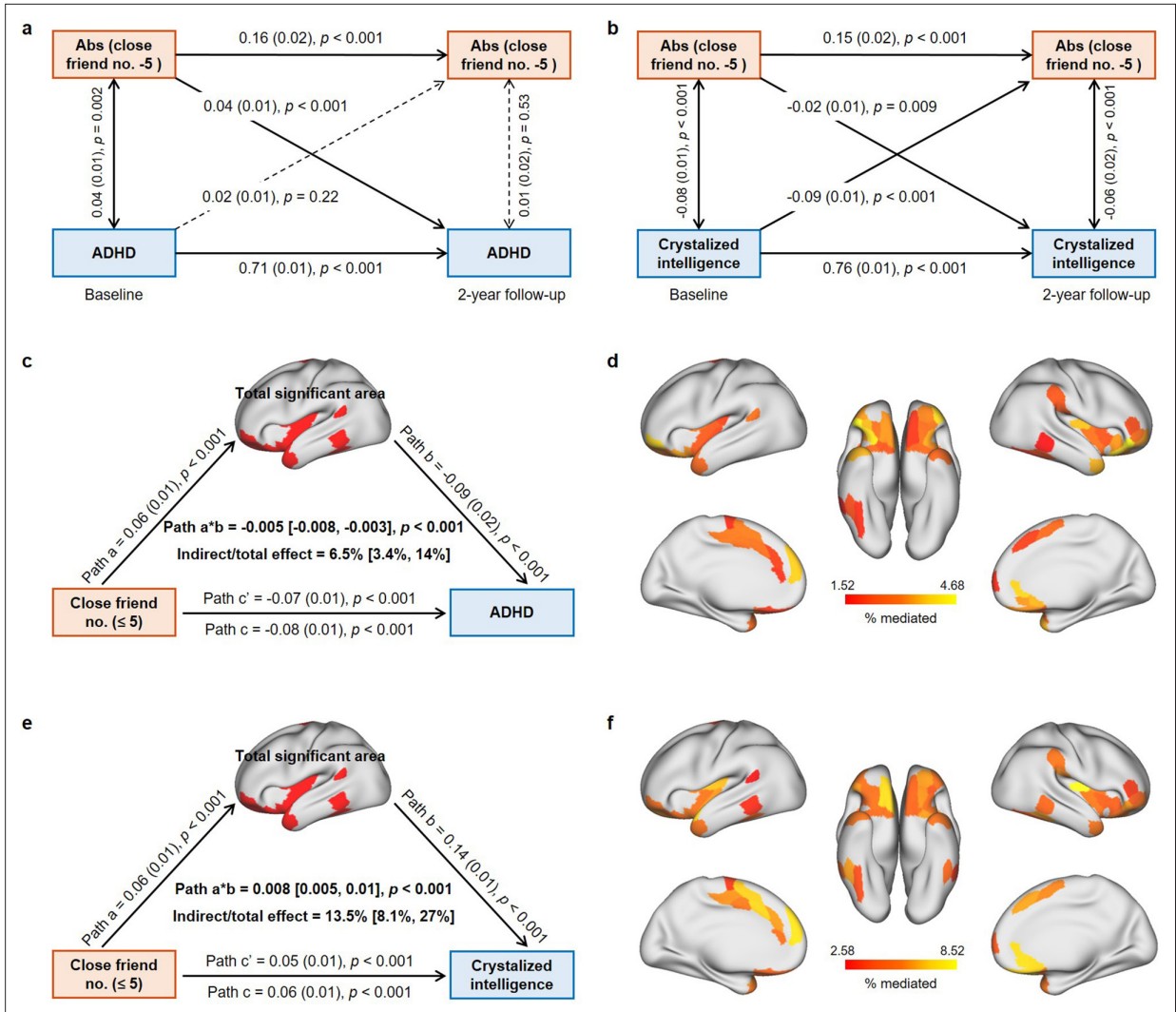

**Figure 5.** Results of longitudinal and mediation analysis in the Adolescent Brain Cognitive Developmental (ABCD) study. (**a**) Cross-lagged panel model (CLPM) of the absolute value of close friendship quantity to 5 and ADHD symptoms (N = 6013). Comparative fit index (CFI) = 0.996, Tucker–Lewis index (TFI) = 0.97, standardized root mean squared residual (SRMR) = 0.002, root mean square error of approximation (RMSEA) = 0.015. (**b**) CLPM of the absolute value of close friendship quantity to 5 and crystalized intelligence (N = 6013). CFI = 0.994, TFI = 0.96, SRMR = 0.003, RMSEA = 0.025. (**c**) Mediation analysis of close friendship quantity, the total area of significant regions, and ADHD symptoms. (**d**) The effect of individual significant cortical areas that mediated the association between close friendship quantity and ADHD symptoms after FDR correction. (**e**) Mediation analysis of close friendship quantity, the total area of significant regions and crystalized intelligence. (**f**) The effect of individual significant cortical areas that mediated the association between close friendship quantity and crystalized intelligence after FDR correction.

The online version of this article includes the following figure supplement(s) for figure 5:

**Figure supplement 1.** Cross-lagged panel models (CLPMs) of close friend number and Adolescent Brain Cognitive Developmental (ADHD) symptoms, and crystalized intelligence in ≤5 and >5 groups, respectively.

of neurotransmitters were not significant except for the μ-opioid receptor (Spearman's rho = 0.44, Bonferroni corrected $p_{perm}$ = 0.02; *Figure 4a and b*). Transcriptomic analyses showed that OPRM1 (Spearman's rho = 0.45, $p_{perm}$ = 0.001; *Figure 4c*) and OPRK1 (Spearman's rho = 0.46, $p_{perm}$ = 0.002; *Figure 4d*) were highly expressed in regions related to the number of close friends.

## Longitudinal and mediation results

As the nonlinear association between the number of close friends and ADHD symptoms is relatively strong and robust, and for cognitive outcomes, only crystalized intelligence was collected at 2-year follow-up in the ABCD study, we focused on these two measures in longitudinal and mediation

analyses. The cross-lagged panel model (CLPM) revealed that participants having closer to five close friends had fewer ADHD symptoms 2 y later ($\beta$ = 0.04, p<0.001; *Figure 5a*). CLPMs in separate groups confirmed that more close friends contributed to fewer ADHD symptoms in ≤5 group ($\beta$ = –0.04, p=0.003; *Figure 5—figure supplement 1a*), but the effect reversed in the >5 group ($\beta$ = 0.05, p=0.019; *Figure 5—figure supplement 1b*). The relationship between the absolute difference of close friend number to five and crystalized intelligence was bidirectional (*Figure 5b*). Only in the ≤5 group was a significant negative correlation found between crystalized intelligence at baseline and the number of close friends at 2-year follow-up ($\beta$ = –0.06, p=0.001; *Figure 5—figure supplement 1c and d*).

Mediation analyses were used to determine whether and the extent to which the association between the number of close friends, ADHD symptoms, and crystalized intelligence could be explained by the identified cortical areas in the ≤5 group. The total identified cortical area partly mediated the association between the number of close friends and ADHD symptoms (6.5%, 95% CI [3.4%, 14%]; path a*b: –0.005, 95% CI [-0.008,–0.003]; *Figure 5c*), and the mediation effects of individual significant regions ranged from 1.52 to 4.68% (*Figure 5d*). Similarly, the association between the number of close friends and crystalized intelligence was partly mediated by the total identified cortical area (13.5%, 95% CI [8.1%, 27%]; path a*b: 0.008, 95% CI [0.005, 0.01]; *Figure 5e*), ranging from 2.58 to 8.52% for each significant region (*Figure 5f*).

## Findings in an independent social network dataset

Utilizing the social network dataset allowed us to extend findings in the ABCD study, as it is an independent and large dataset, a directed friendship network was generated by nomination, and different measures of mental health and cognition were collected (i.e., well-being and grade point average [GPA]). Three indicators of friendship network size (i.e., outdegree, indegree, and reciprocal degree; *Figure 6*) were significantly related to well-being (indegree: $F$ = 38.63, p=$1.8 \times 10^{-17}$; outdegree: $F$ = 33.55, p=$2.9 \times 10^{-15}$; reciprocal degree: $F$ = 53.87, p=$4.8 \times 10^{-24}$; *Figure 6—figure supplement 1a*) and GPA (indegree: $F$ = 28.08, p=$6.7 \times 10^{-13}$; outdegree: $F$ = 46.66, p=$6.2 \times 10^{-21}$; reciprocal degree: $F$ = 192.65, p=$2.1 \times 10^{-83}$; *Figure 6—figure supplement 1b*). Specifically, for well-being, all linear terms were significant, but only the quadratic term of outdegree was significant after Bonferroni correction ($\beta$ = $–2.9 \times 10^{-4}$, $t$ = –3.67, p=$2.4 \times 10^{-4}$, $\Delta R^2$ = 0.07%; *Figure 6—figure supplement 1c*). For GPA, the quadratic terms of all three indicators were significant, and the greatest effect size was observed in the outdegree ($\beta$ = –0.001, $t$ = –6.02, p=$1.8 \times 10^{-9}$, $\Delta R^2$ = 0.17%; *Figure 6—figure supplement 1d*). The two-lines tests revealed that the positive association of outdegree with well-being and GPA diminished once the outward nomination reached 7 or 8 (*Figure 6—figure supplement 1d*). The results confirmed that friendship network size especially outdegree was nonlinearly related to mental health and cognitive outcomes.

## Discussion

The present study showed that close friendship quantity was associated with better mental health and higher cognitive functions in late childhood, and that the beneficial association diminished or reversed when increasing the number of close friends beyond a moderate level. The results also support the hypothesis that a quadratic association exists between the number of close friends and the areas of social brain regions such as the OFC, the ACC, insula, the anterior temporal cortex, and the TPJ. These regions mediated the nonlinear association between close friendship quantity and behavior. Furthermore, the brain differences related to the number of close friends were correlated with measures of the endogenous opioid involvement of the brain regions.

Social relationships play a double-edged role for mental health. Previous research has primarily focused on the positive aspects of social relationships, while the negative effects have received comparatively less attention (*Song et al., 2021*). In our study, we identified a robust nonlinear association of close friend quantity with various mental health and cognitive outcomes in the ABCD study at baseline and 2-year follow-up, and an independent social network dataset. This result demonstrates the persistence of the findings. The findings are in line with past studies, which showed that too large a social network size or too frequent social contacts were not positively correlated with well-being in adults (*Kushlev et al., 2018*; *Ren et al., 2022*; *Stavrova and Ren, 2021*) and were even negatively

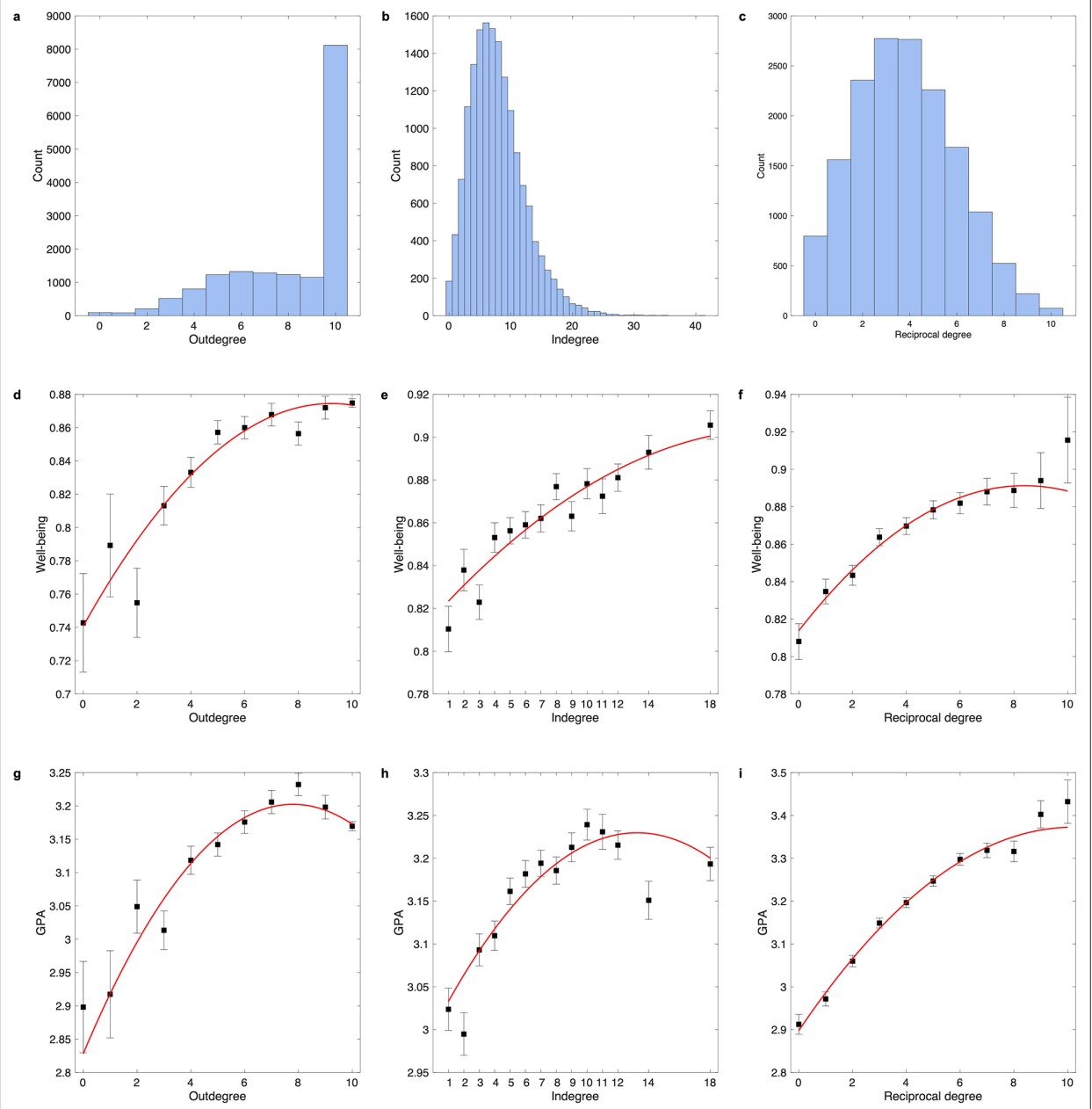

**Figure 6.** Distribution of outdegree, indegree, and reciprocal degree in the social network dataset. (**a**) Distribution of outdegree which is the number of outward nominations. (**b**) Distribution of indegree which is the number of inward nominations. (**c**) Distribution of reciprocal degree which is the number of reciprocal nominations. Relationship of well-being with outdegree (**d**), indegree (**e**), and reciprocal degree (**f**). Relationship of grade point average (GPA) with outdegree (**g**), indegree (**h**), and reciprocal degree (**i**). In each bin, the mean (i.e., black dot) and standard error (i.e., error bar) of the dependent variable are shown. The red line is the fitted quadratic model. For outdegree, sample sizes of bins 1-11 are 92, 87, 208, 519, 803, 1230, 1326, 1286, 1235, 1151 and 8119. For indegree, sample sizes of bins 1-14 are 617, 728, 1116, 1341, 1526, 1563, 1532, 1462, 1273, 1095, 870, 1281, 716 and 936. For reciprocal degree, sample sizes of bins 1-11 are 797, 1561, 2356, 2774, 2765, 2260, 1685, 1039, 524, 220 and 75.

The online version of this article includes the following figure supplement(s) for figure 6:

**Figure supplement 1.** Results of nonlinear association analysis in the social network dataset.

correlated with mental health in adolescents (**Falci and McNeely, 2009**). One explanation is that an individual's cognitive capacity and time limit the size of the social network that an individual can effectively maintain (**Dunbar, 2018**). People devote about 40% of their total social efforts (e.g., time and emotional capital) to just their five most important people (**Bzdok and Dunbar, 2020**). In a phone-call dataset of almost 35 million users and 6 billion calls, a layered structure was found with the innermost

layer having an average of 4.1 people (*Mac Carron et al., 2016*). There is a trade-off between the quantity and quality of friendships, with an increased number of close friends potentially leading to less intimacy. Meanwhile, spending too much time on social activities may lead to insufficient time for study and thereby to lower academic performance. Second, adolescents are particularly susceptible to peer influence (*Berndt, 1979*). Researchers have found that the presence of a peer may increase risk-taking behaviors that can be detrimental to mental health (*Chein et al., 2011*) and reduce cognitive performance (*Wolf et al., 2015*). Having more close friends may increase the possibility of this kind of influence.

Our study revealed a significant link between the number of close friends and the cortical areas of social brain regions in the largest sample of children to date. Studies suggest that two major systems in the brain related to social behavior include the affective system of the ACC, the anterior insula, and the OFC, and the mentalizing system typically involving the TPJ (*Güroğlu, 2022*; *Schmälzle et al., 2017*). The dorsal ACC and anterior insula play an important role in social pain (i.e., painful feelings associated with social disconnection) (*Eisenberger, 2012*). The OFC receives information about socially relevant stimuli such as face expression and gesture from the cortex in the superior temporal sulcus (*Hasselmo et al., 1989a*; *Pitcher and Ungerleider, 2021*), and is involved in social behavior by representing social stimuli in terms of their reward value (*Rolls, 2019b*; *Rolls, 2019a*; *Rolls et al., 2006*). The volume of the OFC is associated with social network size, partly mediated by mentalizing competence (*Powell et al., 2012*). Previous meta-analysis studies report an overlap in brain activation between all mentalizing tasks in the mPFC and posterior TPJ (*Schurz et al., 2014*). Notably, in our study, the positive relationship at the brain level only held for the children with no more than approximately five close friends, which is consistent with the behavioral findings. Furthermore, in these children, the areas of social brain regions partly mediated the relationship of the close friend quantity with ADHD symptoms and crystalized intelligence. Research also indicates that the brain regions regulating social behavior undergo structural development during adolescence (*Blakemore, 2008*; *Lamblin et al., 2017*; *Mills et al., 2014*). Animal studies provide evidence for the causal effect of social relationships on brain development. For instance, adolescent rodents with deprivation of peer contacts showed brain level changes including reduced synaptic pruning in the prefrontal cortex (*Orben et al., 2020*).

Moreover, the brain associative pattern of close friend quantity in children with no more than five close friends was correlated with the density of the µ-opioid receptor, as well as the expression of OPRM1 and OPRK1 genes. It is known that the endogenous opioid system has a vital role in social affiliative processes (*Machin and Dunbar, 2011*). Positron emission tomography studies in human revealed that µ-opioid receptor regulation in brain regions such as the amygdala, anterior insula, and the ACC may preserve and promote emotional well-being in the social environment (*Hsu et al., 2013*). Variation in the OPRM1 gene was associated with individual differences in rejection sensitivity, which was mediated by dorsal ACC activity in social rejection (*Way et al., 2009*). OPRM1 variation was also related to social hedonic capacity (*Troisi et al., 2011*). Pain tolerance, which is associated with activation of the µ-opioid receptor, was correlated with social network size in humans (*Johnson and Dunbar, 2016*). Social behaviors like social laughter and social touch increase pleasurable sensations and triggered endogenous opioid release to maintain social relationships (*Dunbar, 2010*; *Manninen et al., 2017*; *Nummenmaa et al., 2016*). Additionally, the opioid system has found to be associated with major psychiatric disorders especially depression (*Peciña et al., 2019*), which may help explain the association between social relationships and mental health problems.

Several issues should be taken into account when considering our findings. First, as an association study, no causal conclusion should be made in this study. It is unclear whether the number of close friends drives the social brain development or whether children with larger social brains tend to have more close friends. A bidirectional relationship has been reported in the literature (*Dunbar and Shultz, 2007*). Second, it is worth noting that the measures used in the ABCD study and the social network dataset differed, and the breakpoints identified in each dataset were not equivalent. However, relative to the optimal number of close friends, the primary objective of the current study was to examine the nonlinear relationship between the number of close friend and different behavioral outcomes and brain structure. In this sense, the findings from both datasets were similar, and the social network dataset provided valuable information regarding friendship measures and objective cognitive index that extended the results obtained from the ABCD study. Third, the quality of close

friendships was not considered in the ABCD study. However, reciprocal degree is an indirect measure of friendship quality, which was found to be linearly associated with well-being and nonlinearly related to the GPA in the social network dataset. It has been reported that the relationship between having more friends and fewer depressive symptoms in adolescence is mediated by a sense of belonging (*Ueno, 2005*). Although current findings on the relative importance of friendship quantity and quality are inconsistent (*Bruine de Bruin et al., 2020*; *Platt et al., 2014*), it is essential for future studies to incorporate measures of close friendship quality and to test the potential interaction between quantity and quality. Finally, the interpretation of this study should be limited to the particular age range of late childhood and early adolescence, as well as Western culture. Further research is needed to explore whether the nonlinear relationship between the number of close friends and mental health and cognition, and the idea of having around five close friends as a breakpoint, can be generalized to other age ranges and cultures.

In conclusion, this study provides new evidence going beyond previous research that a larger number of close friends up to a moderate level in late childhood is associated with better mental health and higher cognitive functions, and that this can be partly explained by the size of the social brain including the OFC and TPJ, and the endogenous opioid system. This study may have implications for targeted friendship interventions in the transition from late childhood to early adolescence.

## Materials and methods
### Participants and behavioral measures
#### The ABCD study
The ABCD study is tracking the brain development and health of a nationally representative sample of children aged 9–11 y from 21 centers throughout the United States (https://abcdstudy.org). Parents' full written informed consent and all children's assent were obtained by each center. Research protocols were approved by the institutional review board of the University of California, San Diego (no. 160091), and the institutional review boards of the 21 data collection sites (*Auchter et al., 2018*). The current study was conducted on the ABCD Data Release 4.0. At baseline, 8835 individuals from 7512 families (6225 [82.9%] with a child, 1252 [16.7%] with two children, 34 [0.5%] with three children, and 1 [0.01%] with four children) had complete behavioral and structural MRI data. To avoid the influence of family relatedness, we randomly picked only one child in each family, finally resulting in 7512 children, of whom 4290 had 2-year follow-up data.

Close friendships are characterized by enjoying spending time together, having fun, and trust. Participants were asked how many close friends that are boys and girls they have, respectively. Mental health problems were rated by the parent using the Child Behavior Checklist (CBCL), which contains 20 empirically based subscales spanning emotional, social and behavioral domains in subjects aged 6–18 (*Achenbach and Rescorla, 2001*). The CBCL has high inter-interviewer reliability, test–retest reliability, internal consistency, and criterion validity, and therefore is widely utilized by child psychiatrists, developmental psychologists, and other mental health professionals for clinical and research purposes (*Achenbach et al., 1987*). Raw scores were used in analyses, higher scores indicating more severe problems. Cognitive functions were assessed by the NIH Toolbox (*Luciana et al., 2018*), which has good reliability and validity in children (*Akshoomoff et al., 2013*). The toolbox consists of seven different tasks covering episodic memory, executive function, attention, working memory, processing speed, and language abilities, and also provides three composites of crystalized, fluid, and total intelligence (*Weintraub et al., 2013*). Uncorrected standard scores were used in analyses. All 10 cognitive scores were available at baseline, but only crystalized intelligence was collected 2 y later.

#### Social network dataset
In order to extend the findings in the ABCD study, we utilized a publicly available dataset of a social network experiment, conducted among students in 56 middle schools in New Jersey, USA (*Paluck et al., 2016*) (https://www.icpsr.umich.edu/web/civicleads/studies/37070). All parents and students provided informed consent for the survey, and the research protocol was approved by the Princeton University Institutional Review Board. Participants were asked to report which other students (up to 10) in their school they chose to spend time with in the last few weeks, allowing us to generate a directed friendship network. Three kinds of network measures were created for each participant: (1)

outdegree is a measure of sociability and refers to the number of friendship nominations that a participant made to other participants, (2) indegree is a measure of popularity and refers to the number of friendship nominations received from others, and (3) reciprocal degree refers to the number of outward nominations that are reciprocated by an inward nomination from the same person and to some extent reflects the quality of friendship. Well-being was assessed by three questions: 'I feel like I belong at this school,' 'I have stayed home from school because of problems with other students,' and 'During the past month, I have often been bothered by feeling sad and down' (*Ren et al., 2022*). Cognitive function was indirectly measured by the GPA on a 4.0 scale, obtained from school administrative records.

## Structural MRI data

In the ABCD study, 3D T1- and T2-weighted structural images were collected using 3T scanners at 21 data collecting sites (*Casey et al., 2018*). The detailed preprocessing pipeline has been described elsewhere (*Gong et al., 2021*). In brief, we used FreeSurfer v6.0 to preprocess the minimal preprocessed T1- and T2-weighted images downloaded from the ABCD study, including cortical surface reconstruction, subcortical segmentation, smoothed by a Gaussian kernel (FWHM = 10 mm), and estimation of morphometric measures (i.e., cortical area, thickness, and volume). Then, the cortical surface of each subject was registered to a standard fsaverage space and parcellated into 180 cortical regions per hemisphere as defined in the Human Connectome Project multimodal parcellation (HCP-MMP) atlas (*Glasser et al., 2016*). Volumetric reconstructions of subcortical structures were also obtained based on the Aseg atlas (*Fischl et al., 2002*).

## Neurochemical data

Fourteen receptors and transporters across eight different neurotransmitter systems (serotonin: 5HT1a, 5HT1b, 5HT2a, 5HT4, and 5HTT; dopamine: D1, D2, and DAT; GABA: GABAa; glutamate: mGluR5; norepinephrine: NAT; cannabinoid: CB1; opioid: MOR; acetylcholine: VAChT) were investigated. Density estimates were derived from average group maps of healthy volunteers scanned in prior PET and SPECT studies (*Supplementary file 1*). All density maps were downloaded online (https://github.com/juryxy/JuSpace/tree/JuSpace_v1.3/JuSpace_v1.3/PETatlas; *Dukart et al., 2021*), which had been registered and normalized into the Montreal Neurological Institute (MNI) space, and linearly rescaled to 0–100 (*Dukart et al., 2021*). For comparability, the HCP atlas in fsaverage space was converted to individual surface space ('mri_surf2surf') of the MNI brain template ch2 (*Rorden and Brett, 2000*) which was preprocessed by Freesurfer ('recon-all'), and then was projected to volume ('mri_label2vol'). The density maps were parcellated into the 360 cortical regions as the structural MRI data according to the volume-based HCP-MMP atlas. Specifically, for the μ-opioid receptor, occipital cortex served as the reference region (*Kantonen et al., 2020*) and was therefore excluded in analysis.

## Transcriptomic data

Gene expression data was from six neurotypical adult brains in the Allen Human Brain Atlas (*Hawrylycz et al., 2012*). We focused on the opioid receptor genes (i.e., OPRM1 and OPRK1). The preprocessed transcriptomic data were imported from *Arnatkeviciute et al., 2019* (https://doi.org/10.6084/m9.figshare.6852911), including probe-to-gene re-annotation, intensity-based data filtering, and probe selection using RNA-seq data as a reference. Then, samples were assigned to brain regions according to the volume-based HCP-MMP atlas, and expression values were averaged within each region. Since right hemisphere data were only available for two donors, analyses were conducted on the left hemisphere only, finally resulting in 177 brain regions.

## Statistical analysis

### Nonlinear association analysis

The nonlinear associations of close friendship quantity with mental health, cognition, and brain structure were investigated. The quantity of close friendship was log-transformed [$\log_{10}(x + 1)$] in analyses as it has a skewed distribution (*Hobbs et al., 2016*). Two different analytic approaches were used to robustly evaluate the nonlinear relationships. First, we fitted a quadratic regression model ($y = bx^2 + ax + c$) with close friendship quantity as the independent variable. Close friendship quantity was

mean-centered to ensure that the linear (a) and quadratic (b) terms were orthogonal. Three statistical parameters were of interest: a total $F$-value of linear and quadratic terms, reflecting the association between close friendship quantity and the measure of interest (*Li et al., 2022*); the quadratic term, indicating the presence of a nonlinear association; and the linear term. The effect size of the quadratic term was calculated by the change in the overall proportion of variance (adjusted $R^2$) between the quadratic model and the corresponding linear model, and the effect size of the linear term was the $\Delta R^2$ between the linear model and the model with only covariates. The model fits of quadratic and linear models were compared by ANOVA. Although quadratic regression is widely used in psychosocial studies to detect the presence of nonlinearity (*Nook et al., 2018*; *Ren et al., 2022*), simulation studies showed that this approach for testing a U-shaped effect has a high false positive rate (*Simonsohn, 2018*). Therefore, we conducted a two-lines test (*Simonsohn, 2018*) once a significant quadratic term was found, which could estimate a data-driven breakpoint. We then split the data accordingly to fit two linear models, respectively. If the segment slopes have opposite signs and both of them are significant, a U-shaped relation exists. Same analytic approaches were used in behavioral and neuroimaging analyses. In the ABCD study, sex, age, parent education level, household income, ethnicity, puberty, body mass index (BMI), and site were used as covariates of no interest for the behavioral analyses. For the neuroimaging analyses, we additionally controlled for handedness, head motion, and MRI manufacturer. In the social network dataset, we controlled for sex, age, grade, whether the subject was or was not new to the school, and whether or not most friends went to this school. Bonferroni correction was used in behavioral analyses, and FDR correction was used in neuroimaging analyses.

Several sensitivity analyses were performed. To examine the potential sex influence, we conducted nonlinear association analyses in male and female, respectively. The effect of the sex of close friends was tested by separating close friends into same-sex and opposite-sex ones. To validate the findings from data at baseline and to test the hypothesis of stationarity for cross-lagged panel models (CLPM), we replicated the same analyses using the cross-sectional data collected at 2 y later. For neuroimaging analyses, if significant nonlinear associations were detected, we also conducted linear regression models in two groups split by the average breakpoint, respectively.

## Spatial correlation with neurotransmitter density and gene expression

Unthresholded $t$-statistic maps of brain structure associated with close friendship quantity in two groups (i.e., split by the average breakpoint) were used to correlate with neurotransmitter density and gene expression level by Spearman's rank correlation. Bootstrapping was performed to ensure the robustness, and the significance was tested by 5000 times permutation, in which the correlation was re-computed using null $t$-statistic maps obtained by label shuffling for close friendship quantity (*Chen et al., 2021*).

## Cross-lagged panel analysis

Longitudinal relationships of close friendship quantity with ADHD symptoms (i.e., cbcl_scr_syn_attention) and crystalized intelligence were investigated using classic two-wave CLPMs implemented by Mplus 7.0. Firstly, we conducted CLPMs using the absolute value of the difference between close friendship quantity and the breakpoint, and then established CLPMs for participants with the quantity of close friendship ≤breakpoint and >breakpoint at baseline, respectively. We controlled for several stable (i.e., sex, parent education level, ethnicity, and site) and time-variant variables (i.e., age, household income, and puberty) in these models. The CLPMs met important assumptions such as synchronicity and stationarity (*Baribeau et al., 2022*; *Kenny, 1975*). The model parameters were estimated by the full information maximum likelihood method (*Muthén et al., 1987*). The model fit was evaluated by the Tucker–Lewis index (TLI), comparative fit index (CFI), root mean square error of approximation (RMSEA), and standardized root mean squared residual (SRMR), and interpreted using common thresholds of good fit (*Hu and Bentler, 1999*). All CLPMs reported in the current study have a good model fit.

## Mediation analysis

We used the baseline data in the ABCD study to test the associations between close friendship quantity, ADHD symptoms, crystalized intelligence, and brain structure. The total area of the significant brain regions was used as the mediator. Variables were normalized and then entered into the model.

Sex, age, parent education level, household income, ethnicity, pubertal status, BMI, handedness, head motion, MRI manufacturer, and site were used as covariates of no interest. In addition to the total area, the mediation effect of individual significant regions was also evaluated, the p-values of the mediation effect corrected by FDR correction. Total, direct, and indirect associations were estimated by bootstrapping 10,000 times, and the 95% bias-corrected and accelerated confidence interval (CI) was reported. Analyses were performed using the R mediation package.

## Acknowledgements

We thank the children and families whose ongoing participation made this study possible. Data used in the preparation of this article were obtained from the Adolescent Brain Cognitive Development (ABCD) study (https://abcdstudy.org), held in the NIMH Data Archive (NDA). This is a multisite, longitudinal study designed to recruit more than 10,000 children age 9–10 and follow them over 10 years into early adulthood. The ABCD Study is supported by the National Institutes of Health and additional federal partners under award numbers U01DA041048, U01DA050989, U01DA051016, U01DA041022, U01DA051018, U01DA051037, U01DA050987, U01DA041174, U01DA041106, U01DA041117, U01DA041028, U01DA041134, U01DA050988, U01DA051039, U01DA041156, U01DA041025, U01DA041120, U01DA051038, U01DA041148, U01DA041093, U01DA041089, U24DA041123, U24DA041147. A full list of supporters is available at https://abcdstudy.org/federal-partners.html. A listing of participating sites and a complete listing of the study investigators can be found at https://abcdstudy.org/consortium_members/. ABCD consortium investigators designed and implemented the study and/or provided data but did not necessarily participate in the analysis or writing of this report. This manuscript reflects the views of the authors and may not reflect the opinions or views of the NIH or ABCD consortium investigators. Chun Shen is supported by grants from the National Natural Science Foundation of China (no. 82101617) and the China Postdoctoral Science Foundation (no. 2022M710765). Wei Cheng is supported by grants from the National Natural Science Foundation of China (no. 82071997) and the Shanghai Rising-Star Program (no. 21QA1408700). Jianfeng Feng is supported by National Key R&D Program of China (no. 2018YFC1312904 and no. 2019YFA0709502), Shanghai Municipal Science and Technology Major Project (no. 2018SHZDZX01), ZJ Lab, Shanghai Center for Brain Science and Brain-Inspired Technology, and the 111 Project (no. B18015). The funders had no role in study design, data collection, and interpretation, or the decision to submit the work for publication.

## Additional information

### Funding

| Funder | Grant reference number | Author |
| --- | --- | --- |
| National Natural Science Foundation of China | 82101617 | Chun Shen |
| China Postdoctoral Science Foundation | 2022M710765 | Chun Shen |
| National Natural Science Foundation of China | 82071997 | Wei Cheng |
| Shanghai Rising-Star Program | 21QA1408700 | Wei Cheng |
| National Key Research and Development Program of China | 2018YFC1312904 | Jianfeng Feng |
| National Key Research and Development Program of China | 2019YFA0709502 | Jianfeng Feng |
| Shanghai Municipal Science and Technology Major Project | 2018SHZDZX01 | Jianfeng Feng |

| Funder | Grant reference number | Author |
|---|---|---|
| 111 Project | B18015 | Jianfeng Feng |
| ZJ Lab | | Jianfeng Feng |
| Shanghai Center for Brain Science and Brain-Inspired Technology | | Jianfeng Feng |

The funders had no role in study design, data collection and interpretation, or the decision to submit the work for publication.

## Author contributions
Chun Shen, Formal analysis, Funding acquisition, Visualization, Methodology, Writing – original draft, Writing – review and editing; Edmund T Rolls, Conceptualization, Supervision, Validation, Methodology, Writing – review and editing; Shitong Xiang, Data curation, Formal analysis; Christelle Langley, Barbara J Sahakian, Writing – review and editing; Wei Cheng, Supervision, Funding acquisition, Visualization, Methodology, Writing – review and editing; Jianfeng Feng, Conceptualization, Resources, Supervision, Funding acquisition, Writing – review and editing

## Author ORCIDs
Chun Shen  http://orcid.org/0000-0003-1034-0343
Edmund T Rolls  http://orcid.org/0000-0003-3025-1292
Shitong Xiang  http://orcid.org/0000-0002-5513-4714
Jianfeng Feng  http://orcid.org/0000-0002-8890-8288

## Ethics
In the ABCD study, parents' full written informed consent and all children's assent were obtained by each center. Research protocols were approved by the institutional review board of the University of California, San Diego (No.160091), and the institutional review boards of the 21 data collection sites (Auchter et al., 2018). In the social network dataset, all parents and students provided informed consent for the survey, and the research protocol was approved by the Princeton University Institutional Review Board.

## Decision letter and Author response
Decision letter https://doi.org/10.7554/eLife.84072.sa1
Author response https://doi.org/10.7554/eLife.84072.sa2

# Additional files

## Supplementary files
• MDAR checklist
• Supplementary file 1. Detailed information of neurotransmitter density data.

## Data availability
The ABCD dataset is administered by the National Institutes of Mental Health Data Archive and is freely available to all qualified researchers upon submission of an access request (https://nda.nih.gov/abcd). All relevant instructions to obtain the data can be found in https://nda.nih.gov/abcd/request-access. The request is valid for one year. Data use should be in line with the NDA Data Use Certification. The social network dataset is openly available (https://www.icpsr.umich.edu/web/civicleads/studies/37070). Neurochemical data used in analysis are openly available (https://github.com/juryxy/JuSpace/tree/JuSpace_v1.3/JuSpace_v1.3/PETatlas). Gene expression data are openly available (https://figshare.com/articles/dataset/AHBAdata/6852911). The code used for this study is publicly available at https://github.com/chunshen617/friendship (copy archived at *Shen, 2023*).

The following previously published datasets were used:

| Author(s) | Year | Dataset title | Dataset URL | Database and Identifier |
|-----------|------|---------------|-------------|-------------------------|
| Paluck EL, Shepherd HR, Aronow P | 2020 | Changing Climates of Conflict: A Social Network Experiment in 56 Schools, New Jersey, 2012-2013 | https://doi.org/10.3886/ICPSR37070.v2 | CivicLEADS, 10.3886/ICPSR37070.v2 |
| Arnatkevičiūtė A, Fulcher BD, Fornito A | 2019 | A practical guide to linking brain-wide gene expression and neuroimaging data | https://doi.org/10.6084/m9.figshare.6852911 | figshare, 10.6084/m9.figshare.6852911 |

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
