## [Editor Report]

The findings of this study yield important new insights into the relationship between the number of close friends and mental health, cognition, and brain structure. Due to the large sample sizes, the evidence is solid but would have been improved if both of the analysed datasets contained more closely matched measures. This work advances our understanding of how the friendship network relates to young adolescents' mental well-being and cognitive functioning and their underlying neural mechanisms.

---

## [Decision Letter]

**Decision letter after peer review:**

Thank you for submitting your article "Brain and molecular mechanisms underlying the nonlinear association between close friendships, mental health, and cognition in children" for consideration by *eLife*. Your article has been reviewed by 2 peer reviewers, including Robert Whelan as the Reviewing Editor and Reviewer #1, and the evaluation has been overseen by Christian Büchel as the Senior Editor. The following individual involved in the review of your submission has agreed to reveal their identity: Lisa Schreuders (Reviewer #2).

Essential revisions:

1. Confirm that the statistical assumptions needed for CLPMs have been met?

2. In places, the phrasing is such that the reader could infer a direction of the effect in the cross-sectional results (e.g., line 173: "The ideal number of close friends was 5 and the closer to that number, the better *for* participants' mental health." [my emphasis]). Any such phrasing should be amended.

3. Can you put effect sizes more into context for the reader (i.e., what do results of these magnitudes mean in the real world)?

4. Provide a rationale for 5 friends as the breakpoint when the curve appears u-shaped. Looking at Figure 2, there is a grouping of close friend size between ~2 to ~10 (i.e., the bottom of the u is quite flat there). Although the breakpoint is 5 close friends, could one argue that poorer outcomes tend to occur at a higher number of close friends? That is, problems start when the slope of the upward curve starts to steepen (this would also suggest a result more closely aligned with the social network dataset, albeit the measures are different)?

5. Include some more information on the specific age-range that you are interested in for the study in the introduction/discussion.

6. In lines 181-184 it is stated that "Finally, same analyses were performed using the cross-sectional data collected at 2 years later, and the nonlinear associations of the number of close friends with ADHD symptoms and crystalized intelligence remained significant, with an average breakpoint of 4.83 {plus minus} 0.75 close friends (Figure S5)."

6a: So, all other found relations were not significant at follow up two years later? Please state this explicitly.

6b: In the discussion the authors do not elaborate on this result. What do the authors make from this?

7. Re. Figure S10e (explained in lines 230-232). Could the authors please explain how a correlation can be estimated between "the differences in cortical area related to the number of close friends in the two groups". This reads as if a correlation has been performed to test for between-subject differences.

8. It is unclear what the authors mean with the three indicators of friendship network size from the independent social network dataset (indegree, outdegree, reciprocal degree). This makes it hard to interpret the results. Please explicitly explain these terms.

9. Were the independent behavioral variables collected in the ABCD and independent dataset reliable?

*Reviewer #1 (Recommendations for the authors):*

Table 1 in the main manuscript described several characteristics of the ABCD dataset. More information would be helpful for the reader. Could you give a breakdown on all variables for the 2 groups (on the breakpoint of over and under 5 friends)? Are other, potentially relevant, variables available for the two groups? For example, position in family, family size, urban vs. rural dwelling etc. Perhaps rural dwellers have fewer opportunities to have a large circle of close friends, and perhaps rural dwellers are systematically different in other ways (e.g., diet, mental health).

I have little expertise with cross-lagged panel models; can you confirm that the statistical assumptions needed for CLPMs have been met?

I didn't find the inclusion of the second dataset especially convincing. I appreciate that ABCD is unique (and therefore finding a replication sample is very challenging), but the social network dataset was quite different in the measures used. After reading the manuscript I was left with the impression that calling the social network dataset results a 'validation' was somewhat of a stretch. The results of two datasets were broadly similar (breakpoint at 5 in ABCD, ~8 outward nomination in the social network dataset) rather than a validation per se.

In places, the phrasing is such that the reader could infer a direction of the effect in the cross-sectional results (e.g., line 173: "The ideal number of close friends was 5 and the closer to that number, the better *for* participants' mental health." [my emphasis]).

This is not a limitation of your study but, as I mentioned in the public review, the effect sizes are very small. For example, in the social network dataset the β for GPA was 0.001. I would rather read a paper with a huge sample size and small effect sizes than vice-versa but could you put the effect sizes into context for the reader?

Looking at Figure 2, there is a grouping of close friend size between ~2 to ~10 (i.e., the bottom of the u is quite flat there). Although the breakpoint is 5 close friends, could one argue that poorer outcomes tend to occur at a higher number of close friends? That is, problems start when the slope of the upward curve starts to steepen (this would also suggest a result more closely aligned with the social network dataset, albeit the measures are different)? The discussion of 5 as the breakpoint is more appropriate when the curve is v-shaped rather than u-shaped.

*Reviewer #2 (Recommendations for the authors):*

I am excited about this paper and believe it is of added value to the existing literature, but I would appreciate a clearer theoretical framework with regards to why certain biological constructs were assessed in this study (introduction) and interpretation of the results in the context of the existing literature (discussion) (see public review). I outline my remaining questions in more detail below.

The authors do not specify anything about the specific age-range they were interested in for the study in the introduction/discussion.

In lines 181-184 it is stated that "Finally, same analyses were performed using the cross-sectional data collected at 2 years later, and the nonlinear associations of the number of close friends with ADHD symptoms and crystalized intelligence remained significant, with an average breakpoint of 4.83 {plus minus} 0.75 close friends (Figure S5)."

(a) So, all other found relations were not significant at follow up two years later? Please state this explicitly.

(b) Furthermore, I believe in the discussion the authors do not elaborate on this result. What do the authors make from this?

I do not follow what is exactly being tested in Figure S10e (explained in lines 230-232). Could the authors please explain how a correlation can be estimated between "the differences in cortical area related to the number of close friends in the two groups". This reads as if a correlation has been performed to test for between-subject differences.

It is unclear what the authors mean with the three indicators of friendship network size from the independent social network dataset (indegree, outdegree, reciprocal degree). This makes it hard to interpret the results. Please explicitly explain these terms.

Were the independent behavioral variables collected in the ABCD and independent dataset reliable?

---

## [Author Response]

Essential revisions:1. Confirm that the statistical assumptions needed for CLPMs have been met?

Cross-lagged panel model (CLPM) is a method to examine the directional effects between two variables over time, simultaneously considering the synchronous and autoregressive relations. In the present study, we used the most basic CLPM which includes two constructs measured at two time points. Several important assumptions need to be met for CLPM (Baribeau et al., 2022; Kenny, 1975).

1) The synchronicity assumption requires that all measurements at each time point occurred at the same time, which is met for the data collection in the ABCD study.

2) Stationarity requires variables and relationships stay the same across time. We have confirmed that ADHD symptoms and crystalized intelligence were nonlinearly related to the number of close friends, with a breakpoint of around 5 close friends, at both baseline and 2-year follow-up (see the Results section).

3) Measurement error: The variables are assumed to be measured without error. However, it is difficult to assess measurement error directly. The assessments of ADHD symptoms and crystalized intelligence in the ABCD study are reliable and valid, which is helpful to minimize the measurement error.

4) Linearity: Relations between variables of interest are linear. We fitted CLPMs using the absolute difference of close friend quantity and the breakpoint, and CLPMs in participants with ≤5 and >5 close friends, respectively. Although we found nonlinear relationships of the number of close friends with ADHD symptoms and crystalized intelligence, the associations within groups by the breakpoint is linear.

5) A sufficient sample size of 200 participants has been proposed for CLPMs with two waves. Indeed, our study exceeds this, our sample size was in the thousands.

Finally, we used the Tucker-Lewis Index (TLI), Comparative Fit Index (CFI), root mean square error of approximation (RMSEA) and standardized root mean squared residual (SRMR) to assess the model fit. For the maximum likelihood method, a cutoff value close to 0.95 for TFI and CFI, a cutoff value close to 0.06 for RMSEA and 0.08 for SRMR indicate a good fit (Hu & Bentler, 1999). All CLPMs conducted in the present study have a good model fit.

We have now stated the assumptions and model fit criteria in the Statistical analysis, and reported the values of the model fit indices in the caption of Figure 5.

“The CLPMs met important assumptions such as synchronicity and stationarity (Baribeau et al., 2022; Kenny, 1975). The model parameters were estimated by the full information maximum likelihood method (Muthén et al., 1987). The model fit was evaluated by the Tucker-Lewis Index (TLI), Comparative Fit Index (CFI), root mean square error of approximation (RMSEA) and standardized root mean squared residual (SRMR), and interpreted using common thresholds of good fit (Hu & Bentler, 1999). All CLPMs reported in the current study have a good model fit.” (Lines 552-559, Page 27)

“a. CLPM of the absolute value of close friendship quantity to 5 and ADHD symptoms (N=6,013). CFI = 0.996, TFI = 0.97, SRMR = 0.002, RMSEA = 0.015. b. CLPM of the absolute value of close friendship quantity to 5 and crystalized intelligence (N=6,013). CFI = 0.994, TFI = 0.96, SRMR = 0.003, RMSEA = 0.025.”

2. In places, the phrasing is such that the reader could infer a direction of the effect in the cross-sectional results (e.g., line 173: "The ideal number of close friends was 5 and the closer to that number, the better *for* participants' mental health." [my emphasis]). Any such phrasing should be amended.

Thank you for pointing out this. Indeed, it is impossible to infer a direction of the effect from cross-sectional data, as such data do not allow for the assessment of temporal precedence between variables. Further revisions shown in the paper:

“Both mental health and cognition were positively associated with close friend quantity, with an ideal number of around 5.” (see Lines 171-173, Page 9)

3. Can you put effect sizes more into context for the reader (i.e., what do results of these magnitudes mean in the real world)?

We agree with the Reviewer that the large sample size derives low P-values for even small effect sizes. Therefore, we reported effect sizes of the quadratic and linear terms calculated by the change in the adjusted R^2^. For example, at baseline, the greatest effect sizes of quadratic terms were observed for social problems and total intelligence. The results showed that the quadratic term of close friend quantity additionally explained 0.43% and 0.50% of the variability, respectively, compared with the corresponding linear models. Complex psychological factors are typically determined by a multitude of causes, and any individual cause is likely to have only a small effect (Götz et al., 2022). Although the effect size is small, our primary aim is to investigate the nonlinear links between the number of close friend, mental health, and cognition, as the effect of the friendship network size at the high end remains largely unexplored in children and adolescent. We utilized two independent datasets of large sample sizes and different statistical approaches to show that the nonlinear associations are reliable and solid to various confounders. As the friendship quantity is a modifiable factor, a small but reliable effect size can be valuable.

To further contextualize the effect sizes, we have included the following in the Results:

“For mental health, the greatest effect sizes of the quadratic terms were observed for social problems (β = 0.08, t = 5.92, p = 3.3⨉10^-9^, ΔR^2^ = 0.43%) and attention problems (β = 0.12, t = 5.83, p = 5.8⨉10^-9^, ΔR^2^ = 0.42%), suggesting that the quadratic term of close friend quantity additionally explained 0.43% and 0.50% of the variability compared with the corresponding linear model.” (Lines 159-164, Pages 8-9).

4. Provide a rationale for 5 friends as the breakpoint when the curve appears u-shaped. Looking at Figure 2, there is a grouping of close friend size between ~2 to ~10 (i.e., the bottom of the u is quite flat there). Although the breakpoint is 5 close friends, could one argue that poorer outcomes tend to occur at a higher number of close friends? That is, problems start when the slope of the upward curve starts to steepen (this would also suggest a result more closely aligned with the social network dataset, albeit the measures are different)?

In our study, we conducted two-lines tests to detect U-shaped relationships, which involves a sign change, to complement the estimation of quadratic regression models. The Robin Hood algorithm is used to set a breakpoint to maximize the statistical power for U-shape testing instead of fitting the data as well as possible. Thus, the algorithm determines a breakpoint that will increase the statistical strength of the weaker of the two lines by allocating more observations to that segment without overly attenuating its slope (Simonsohn, 2018). We found the breakpoints for mental health and cognition ranged from 4 to 6 close friends, with an average of 4.89 ± 0.68 (Figure 2h).

However, we acknowledge the Reviewer's concern that poorer outcomes may occur at higher numbers of close friends. As we have clarified, the primary objective of our study is to examine the nonlinear relationships between the number of close friends, mental health, and cognition rather than determining the optimal number of close friends. Therefore, we have now revised our statement of the breakpoint to "around 5" or “approximately 5” in the manuscript, and provided additional explanations in the Discussion.

“Second, it is worth noting that the measures used in the ABCD study and the social network dataset differed, and the breakpoints identified in each dataset were not equivalent. However, relative to the optimal number of close friends, the primary objective of the current study was to examine the nonlinear relationship between the number of close friend and different behavioral outcomes and brain structure.” (Lines 364-369, Page 18)

5. Include some more information on the specific age-range that you are interested in for the study in the introduction/discussion.

Thank you for your constructive comment. We have revised the Introduction to make it more focused on the specific age-range of interest by updating the reference and adding the specific age-range when citing it. Additionally, we have clarified the theoretical framework by explaining the rationale behind assessing the social brain.

“For now, only a few empirical studies have examined the nonlinear association between social relationships, mental health, and cognition in children and adolescents. For instance, a large study of a nationally representative sample in the United States reported that adolescents with either too many or too few friends had higher levels of depressive symptoms (Falci & McNeely, 2009).” (Lines 77-81, Pages 4-5)

“Despite a large body of evidence linking friendships to mental health and cognition, we know relatively little about the underlying mechanisms involved (Pfeifer & Allen, 2021). The social brain hypothesis proposes that the evolution of brain size is driven by complex social selection pressures (Dunbar & Shultz, 2007). Animal studies have shown that social network size can predict the volume of the mid-superior temporal sulcus (Sallet et al., 2011; Testard et al., 2022), a region in which neurons respond to socially relevant stimuli such as face expression and head movement to make or break social contact (Hasselmo, Rolls, & Baylis, 1989; Hasselmo, Rolls, Baylis, et al., 1989). In human neuroimaging studies, several key brain regions, including the medial prefrontal cortex (mPFC, i.e. orbitofrontal [OFC] and anterior cingulate cortex [ACC]), the cortex in the superior temporal sulcus (STS), the temporo-parietal junction (TPJ), amygdala, and the anterior insula, have been implicated in social cognitive processes (Frith & Frith, 2007). Moreover, there has been an increasing number of studies dedicated to investigating the social brain in children and adolescents over the past decade (Andrews et al., 2021; Burnett et al., 2011).

At the molecular level, the μ-opioid receptor is widely distributed in the brain, particularly in regions associated with social pain such as the ACC and anterior insula (Baumgärtner et al., 2006). Recent studies have identified the crucial role of μ-opioid receptors in forming and maintaining friendships (Dunbar, 2018), and variations in the μ-opioid receptor gene have been related to individual differences in rejection sensitivity (Way et al., 2009). In addition, other neurotransmitters including dopamine, serotonin, GABA, and noradrenaline may interact with the opioids, and are involved in social affiliation and social behavior (Machin & Dunbar, 2011). Dysregulation of the social brain and neurotransmitter systems is also implicated in the pathophysiology of major psychiatric disorders (Porcelli et al., 2019). Taken together, it is suggested that changes in the social brain might explain the relationship between social connections and mental health (Lamblin et al., 2017). However, the empirical evidence on this topic is limited in late childhood and adolescence.” (Lines 89-117, Pages 5-6)

We have also rewritten the Discussion to make the main messages in each paragraph clear.

“Social relationships play a double-edged role for mental health. Previous research has primarily focused on the positive aspects of social relationships, while the negative effects have received comparatively less attention (Song et al., 2021). In our study, we identified a robust nonlinear association of close friend quantity with various mental health and cognitive outcomes in the ABCD study at baseline and 2-year follow-up, and an independent social network dataset. This result demonstrates the persistence of the findings. The findings are in line with past studies, which showed that too large a social network size or too frequent social contacts were not positively correlated with well-being in adults (Kushlev et al., 2018; Ren et al., 2022; Stavrova & Ren, 2021) and were even negatively correlated with mental health in adolescents (Falci & McNeely, 2009). One explanation is that an individual’s cognitive capacity and time limit the size of the social network that an individual can effectively maintain (Dunbar, 2018). People devote about 40% of their total social efforts (e.g., time and emotional capital) to just their 5 most important people (Bzdok & Dunbar, 2020). In a phone-call dataset of almost 35 million users and 6 billion calls, a layered structure was found with the innermost layer having an average of 4.1 people (Mac Carron et al., 2016). There is a trade-off between the quantity and quality of friendships, with an increased number of close friends potentially leading to less intimacy. Meanwhile, spending too much time on social activities may lead to insufficient time for study and thereby to lower academic performance. Second, adolescents are particularly susceptible to peer influence (Berndt, 1979). Researchers have found that the presence of a peer may increase risk-taking behaviors which can be detrimental to mental health (Chein et al., 2011), and reduce cognitive performance (Wolf et al., 2015). Having more close friends may increase the possibility of this kind of influence.

Our study revealed a significant link between the number of close friends and the cortical areas of social brain regions in the largest sample of children to date. Studies suggest that two major systems in the brain related to social behavior include the affective system of the ACC, the anterior insula, and the OFC, and the mentalizing system typically involving the TPJ (Güroğlu, 2022; Schmälzle et al., 2017). The dorsal ACC and anterior insula play an important role in social pain (i.e., painful feelings associated with social disconnection) (Eisenberger, 2012). The OFC receives information about socially relevant stimuli such as face expression and gesture from the cortex in the superior temporal sulcus (Hasselmo, Rolls, & Baylis, 1989; Pitcher & Ungerleider, 2021), and is involved in social behavior by representing social stimuli in terms of their reward value (Rolls, 2019a, 2019b; Rolls et al., 2006). The volume of the OFC is associated with social network size, partly mediated by mentalizing competence (Powell et al., 2012). Previous meta-analysis studies report an overlap in brain activation between all mentalizing tasks in the mPFC and posterior TPJ (Schurz et al., 2014). Notably, in our study, the positive relationship at the brain level only held for the children with no more than approximately 5 close friends, which is consistent with the behavioral findings. Furthermore, in these children, the areas of social brain regions partly mediated the relationship of the close friend quantity with ADHD symptoms and crystalized intelligence. Research also indicates that the brain regions regulating social behavior undergo structural development during adolescence (Blakemore, 2008; Lamblin et al., 2017; Mills et al., 2014). Animal studies provide evidence for the causal effect of social relationships on brain development. For instance, adolescent rodents with deprivation of peer contacts showed brain level changes including reduced synaptic pruning in the prefrontal cortex (54).

Moreover, the brain associative pattern of close friend quantity in children with no more than five close friends was correlated with the density of the μ-opioid receptor, as well as the expression of OPRM1 and OPRK1 genes. It is known that the endogenous opioid system has a vital role in social affiliative processes (Machin & Dunbar, 2011). Positron emission tomography studies in human revealed that μ-opioid receptor regulation in brain regions such as the amygdala, anterior insula, and the ACC may preserve and promote emotional well-being in the social environment (Hsu et al., 2013). Variation in the OPRM1 gene was associated with individual differences in rejection sensitivity, which was mediated by dorsal ACC activity in social rejection (Way et al., 2009). OPRM1 variation was also related to social hedonic capacity (Troisi et al., 2011). Pain tolerance, which is associated with activation of the μ-opioid receptor, was correlated with social network size in humans (Johnson & Dunbar, 2016). Social behaviors like social laughter and social touch increase pleasurable sensations and triggered endogenous opioid release to maintain social relationships (Dunbar, 2010; Manninen et al., 2017; Nummenmaa et al., 2016). Additionally, the opioid system has found to be associated with major psychiatric disorders especially depression (Peciña et al., 2019), which may help explain the association between social relationships and mental health problems.” (Lines 291-358, Pages 15-18)

6. In lines 181-184 it is stated that "Finally, same analyses were performed using the cross-sectional data collected at 2 years later, and the nonlinear associations of the number of close friends with ADHD symptoms and crystalized intelligence remained significant, with an average breakpoint of 4.83 {plus minus} 0.75 close friends (Figure S5)."6a: So, all other found relations were not significant at follow up two years later? Please state this explicitly.

We have clarified the results of nonlinear association analyses using 2-year follow-up data.

“Finally, the same analyses were performed using the cross-sectional data collected 2 years later (Figure 2—figure supplement 5). The number of close friends was significantly associated with 10 out of 20 mental health measures, and 3 out of 6 cognitive scores. Significant nonlinear associations were observed between close friend quantity and 5 measures, with an average breakpoint of 4.60 ± 0.55 close friends. The greatest effect sizes of the quadratic terms were observed for attention problems (β = 0.10, t = 3.63, p = 2.9⨉10^-4^, ΔR^2^ = 0.27%) and crystalized intelligence (β = -0.24, t = -4.70, p = 2.7⨉10^-6^, ΔR^2^ = 0.36%) for mental health and cognition, respectively.” (Lines 180-187, Pages 9-10)

6b: In the discussion the authors do not elaborate on this result. What do the authors make from this?

There are two reasons to examine the nonlinear relationship between the number of close friends and mental health and cognition in the ABCD study at 2-year follow-up: (1) validate the nonlinear relationship observed at baseline, and (2) investigate the influence of age on the breakpoint of close friend quantity. In addition to show the reliability of nonlinear findings, this analysis confirmed the important hypothesis of stationarity for the cross-lagged panel model, which requires variables and relationships stay the same across time. We have now described the aim of this analysis in the Method and briefly stated the findings in the Discussion.

“To validate the findings from data at baseline and to test the hypothesis of stationarity for cross-lagged panel models (CLPM), we replicated the same analyses using the cross-sectional data collected at 2 years later.” (Lines 529-531, Page 26)

“In our study, we identified a robust nonlinear association of close friend quantity with various mental health and cognitive outcomes in the ABCD study at baseline and 2-year follow-up, and an independent social network dataset. This result demonstrates the persistence of the findings.” (Lines 293-297, Page 15)

7. Re. Figure S10e (explained in lines 230-232). Could the authors please explain how a correlation can be estimated between "the differences in cortical area related to the number of close friends in the two groups". This reads as if a correlation has been performed to test for between-subject differences.

In order to illustrate the patterns of nonlinear associations between the number of close friends and cortical area, we performed linear regression models in participants with ≤ 5 and > 5 close friends, respectively. Then, we examined the spatial correlation between the unthresholded t-statistic maps of two groups (i.e., obtained from linear regression). We found the correlation was not significant (r = -0.02, *p* = 0.78; Figure 3—figure supplement 5e), suggesting that the brain associative patterns of close friend quantity in two groups were not similar. Now we have revised the statement in the Results to avoid confusion.

“Moreover, the cortical area associative patterns of close friend quantity in the two groups were not correlated (r = -0.02, p = 0.78; Figure 3—figure supplement 5e).” (Lines 219-221, Page 11)

8. It is unclear what the authors mean with the three indicators of friendship network size from the independent social network dataset (indegree, outdegree, reciprocal degree). This makes it hard to interpret the results. Please explicitly explain these terms.

We agree with the Reviewer that these terms were not explicitly explained in the manuscript. In the social network dataset, participants were asked to report which other students in their school they chose to spend time with in the last few weeks, allowing us to generate a directed friendship network. Outdegree refers to the number of friendship nominations that a participant made to others (i.e., outward nomination), and is a measure of their sociability. Indegree is a measure of popularity and refers to the number of friendship nominations received from other participants (i.e., inward nomination). Reciprocal degree is defined as the number of outward nominations that are reciprocated by an inward nomination from the same person, to some extent reflecting the quality of friendship. Now we have clarified the definitions in the Methods.

“Participants were asked to report which other students (up to ten) in their school they chose to spend time with in the last few weeks, allowing us to generate a directed friendship network. Three kinds of network measures were created for each participant: (1) outdegree is a measure of sociability and refers to the number of friendship nominations that a participant made to other participants, (2) indegree is a measure of popularity and refers to the number of friendship nominations received from others, and (3) reciprocal degree refers to the number of outward nominations that are reciprocated by an inward nomination from the same person and to some extent reflects the quality of friendship.” (Lines 434-442, Pages 21-22)

9. Were the independent behavioral variables collected in the ABCD and independent dataset reliable?

In the ABCD study, mental health problems were rated by the parent using the Child Behavior Checklist (CBCL). The CBCL has high inter-interviewer reliability (ICC > 0.9), test-retest reliability (r for each scale > 0.8), internal consistency (mean Cronbach’s α for each scale = 0.85) and criterion validity, and therefore is widely utilized by child psychiatrists, developmental psychologists, and other mental health professionals for clinical and research purposes (Achenbach et al., 1987; Achenbach & Rescorla, 2001). Cognitive functions were assessed by the NIH Toolbox (Luciana et al., 2018). The toolbox has showed excellent test-retest reliability (ICC > 0.78), robust developmental effects across the childhood age range, and strong correlations with established measures of similar abilities and school performance (Akshoomoff et al., 2013; Weintraub et al., 2013). In the social network dataset, as no well-documented mental health questionnaire was available, we used three binary questions as Ren et al.’s study (Ren et al., 2022) to measure well-being. For cognition, we obtained the grade point average from school administrative records as an indirect index.

We have cited references to demonstrate the reliability and validity of behavioral variables used in these two datasets, and briefly stated in the Introduction.

“Mental health problems were rated by the parent using the Child Behavior Checklist (CBCL), which contains 20 empirically based subscales spanning emotional, social and behavioral domains in subjects aged 6 to 18 (Achenbach & Rescorla, 2001). The CBCL has high inter-interviewer reliability, test-retest reliability, internal consistency and criterion validity, and therefore is widely utilized by child psychiatrists, developmental psychologists, and other mental health professionals for clinical and research purposes (Achenbach et al., 1987). Raw scores were used in analyses, higher scores indicating more severe problems. Cognitive functions were assessed by the NIH Toolbox (Luciana et al., 2018), which has good reliability and validity in children (Akshoomoff et al., 2013). The toolbox consists of seven different tasks covering episodic memory, executive function, attention, working memory, processing speed, and language abilities, and also provides three composites of crystalized, fluid, and total intelligence (Weintraub et al., 2013).” (Lines 412-424, Pages 20-21)

“Well-being was assessed by three questions: “I feel like I belong at this school”, “I have stayed home from school because of problems with other students”, and “During the past month, I have often been bothered by feeling sad and down” (Ren et al., 2022). Cognitive function was indirectly measured by the GPA on a 4.0 scale, obtained from school administrative records.” (Lines 442-447, Page 22)

“These datasets provided reliable measures of close friend quantity, mental health, and cognition, and included a combined total of more than 23,000 participants (Figure 1a).” (Lines 123-125, Page 7)

Reviewer #1 (Recommendations for the authors):Table 1 in the main manuscript described several characteristics of the ABCD dataset. More information would be helpful for the reader. Could you give a breakdown on all variables for the 2 groups (on the breakpoint of over and under 5 friends)? Are other, potentially relevant, variables available for the two groups? For example, position in family, family size, urban vs. rural dwelling etc. Perhaps rural dwellers have fewer opportunities to have a large circle of close friends, and perhaps rural dwellers are systematically different in other ways (e.g., diet, mental health).

Thank you for the helpful comment. We have now revised Table 1 to show detailed demographic characteristics at baseline in terms of the breakpoint of 5 close friends and the statistical differences between two groups (please see Table 1 below). According to the Reviewer’s suggestion, we also provided information of family size reported by the parent (question: “How many people are living at your address? INCLUDE everyone who is living or staying at your address for more than 2 months.”), and urbanicity by externally linked to census data. The Census Bureau identifies two types of urban areas, including urbanized areas of 50,000 or more people and urban clusters of at least 2500, but less than 50,000 people. In the ABCD study, we found children with no more than 5 close friends have slightly larger family while no difference of urbanicity compared with children with more than 5 close friends.

I have little expertise with cross-lagged panel models; can you confirm that the statistical assumptions needed for CLPMs have been met?

This suggestion has been addressed in the first question in Essential revisions.

I didn't find the inclusion of the second dataset especially convincing. I appreciate that ABCD is unique (and therefore finding a replication sample is very challenging), but the social network dataset was quite different in the measures used. After reading the manuscript I was left with the impression that calling the social network dataset results a 'validation' was somewhat of a stretch. The results of two datasets were broadly similar (breakpoint at 5 in ABCD, ~8 outward nomination in the social network dataset) rather than a validation per se.

We agree with the Reviewer that the social network dataset is not a direct out-of-sample validation as the measures of mental health and cognition are not identical to those in the ABCD due to the data limitation. However, as we stated before, the primary aim of the present study is to investigate the potential nonlinear relationship between close friend quantity and various mental health and cognitive outcomes. In this sense, we think the findings in these two independent datasets were similar. Moreover, the friendship measures obtained from a directed friendship network in the social network datasets are informative and could be a complementary to the one question of close friend quantity used in the ABCD study. Therefore, we think utilizing the social network dataset extends findings in the ABCD study. We have now avoided to use the term of validation and discussed this limitation in the manuscript.

“Second, it is worth noting that the measures used in the ABCD study and the social network dataset differed, and the breakpoints identified in each dataset were not equivalent. However, relative to the optimal number of close friends, the primary objective of the current study was to examine the nonlinear relationship between the number of close friend and different behavioral outcomes and brain structure. In this sense, the findings from both datasets were similar, and the social network dataset provided valuable information regarding friendship measures and objective cognitive index that extended the results obtained from the ABCD study.” (Lines 364-372, Page 18)

In places, the phrasing is such that the reader could infer a direction of the effect in the cross-sectional results (e.g., line 173: "The ideal number of close friends was 5 and the closer to that number, the better *for* participants' mental health." [my emphasis]).

This suggestion has been addressed in the second question in Essential revisions.

This is not a limitation of your study but, as I mentioned in the public review, the effect sizes are very small. For example, in the social network dataset the β for GPA was 0.001. I would rather read a paper with a huge sample size and small effect sizes than vice-versa but could you put the effect sizes into context for the reader?

This suggestion has been addressed in the third question in Essential revisions.

Looking at Figure 2, there is a grouping of close friend size between ~2 to ~10 (i.e., the bottom of the u is quite flat there). Although the breakpoint is 5 close friends, could one argue that poorer outcomes tend to occur at a higher number of close friends? That is, problems start when the slope of the upward curve starts to steepen (this would also suggest a result more closely aligned with the social network dataset, albeit the measures are different)? The discussion of 5 as the breakpoint is more appropriate when the curve is v-shaped rather than u-shaped.

This suggestion has been addressed in the fourth question in Essential revisions.

Reviewer #2 (Recommendations for the authors):I am excited about this paper and believe it is of added value to the existing literature, but I would appreciate a clearer theoretical framework with regards to why certain biological constructs were assessed in this study (introduction) and interpretation of the results in the context of the existing literature (discussion) (see public review). I outline my remaining questions in more detail below.

We thank the Reviewer for the thorough review and very positive evaluation of our study. We have now revised the Introduction to provide a more detailed explanation of the rational for assessing the specific biological constructs, and also expended the Discussion to better contextualize our findings within the existing literature.

The authors do not specify anything about the specific age-range they were interested in for the study in the introduction/discussion.

This suggestion has been addressed in the fifth question in Essential revisions.

In lines 181-184 it is stated that "Finally, same analyses were performed using the cross-sectional data collected at 2 years later, and the nonlinear associations of the number of close friends with ADHD symptoms and crystalized intelligence remained significant, with an average breakpoint of 4.83 {plus minus} 0.75 close friends (Figure S5)."(a) So, all other found relations were not significant at follow up two years later? Please state this explicitly.(b) Furthermore, I believe in the discussion the authors do not elaborate on this result. What do the authors make from this?

This suggestion has been addressed in the sixth question in Essential revisions.

I do not follow what is exactly being tested in Figure S10e (explained in lines 230-232). Could the authors please explain how a correlation can be estimated between "the differences in cortical area related to the number of close friends in the two groups". This reads as if a correlation has been performed to test for between-subject differences.

This suggestion has been addressed in the seventh question in Essential revisions.

It is unclear what the authors mean with the three indicators of friendship network size from the independent social network dataset (indegree, outdegree, reciprocal degree). This makes it hard to interpret the results. Please explicitly explain these terms.

This suggestion has been addressed in the eighth question in Essential revisions.

Were the independent behavioral variables collected in the ABCD and independent dataset reliable?

This suggestion has been addressed in the ninth question in Essential revisions.